# Magnetic resonance measurements of cellular and sub-cellular membrane structures in live and fixed neural tissue

Nathan H Williamson[1†], Rea Ravin[1,2†], Dan Benjamini[1,3], Hellmut Merkle[4], Melanie Falgairolle[4], Michael James O'Donovan[4], Dvir Blivis[4], Dave Ide[4,5], Teddy X Cai[1], Nima S Ghorashi[6], Ruiliang Bai[1,7], Peter J Basser[1*]

[1]Eunice Kennedy Shriver National Institute of Child Health and Human Development, National Institutes of Health, Bethesda, United States; [2]Celoptics, Rockville, United States; [3]Center for Neuroscience and Regenerative Medicine, Henry Jackson Foundation, Bethesda, United States; [4]National Institute of Neurological Disorders and Stroke, National Institutes of Health, Bethesda, United States; [5]National Institute of Mental Health, National Institutes of Health, Bethesda, United States; [6]Cardiovascular Branch, National Heart, Lung, and Blood Institute, National Institutes of Health, Bethesda, United States; [7]Interdisciplinary Institute of Neuroscience and Technology, School of Medicine, Zhejiang University, Hangzhou, China

**Abstract** We develop magnetic resonance (MR) methods for real-time measurement of tissue microstructure and membrane permeability of live and fixed excised neonatal mouse spinal cords. Diffusion and exchange MR measurements are performed using the strong static gradient produced by a single-sided permanent magnet. Using tissue delipidation methods, we show that water diffusion is restricted solely by lipid membranes. Most of the diffusion signal can be assigned to water in tissue which is far from membranes. The remaining 25% can be assigned to water restricted on length scales of roughly a micron or less, near or within membrane structures at the cellular, organelle, and vesicle levels. Diffusion exchange spectroscopy measures water exchanging between membrane structures and free environments at 100 s$^{-1}$.

*For correspondence:
pjbasser@helix.nih.gov

†These authors contributed equally to this work

**Competing interests:** The authors declare that no competing interests exist.

## Introduction

The first diffusion tensor images of brain tissue showed diffusion anisotropy in white matter (*Basser et al., 1994*). It was postulated that this anisotropy is due to myelin membranes and other cellular components impeding water mobility more in the direction perpendicular to the oriented fibers than parallel to them. By process of elimination, *Beaulieu and Allen (1994)* concluded that the origin of diffusion anisotropy in white matter is due to membranes (*Beaulieu, 2002*). New methods to clear lipid membranes while leaving other tissue components intact (*Chung et al., 2013*; *Tainaka et al., 2018*) have confirmed directly that diffusion becomes isotropic and diffusivity approaches the value of free water after complete delipidation (*Leuze et al., 2017*).

A characteristic of MR is that spin magnetization retains the history of motions encoded during the experimental pulse sequence (*Callaghan et al., 2007*). Diffusion MR measures the spin echo (*Hahn, 1950*) signal attenuation of nuclear spins which displace randomly in the presence of a magnetic field gradient (*Carr and Purcell, 1954*; *Callaghan et al., 1991*; *Callaghan, 2011*). The displacements contain averaged information about the hindrances and restrictions which the molecules experienced during their random trajectories through the microstructure. A diffusion encoding time

defines how long the motions are observed. It also defines the length scales of the displacements, albeit implicitly due to the complex scaling between displacements and time for diffusion in structured media (**Novikov et al., 2014**). In conventional diffusion MRI, the diffusion encoding time is held constant and the gradient strength is incrementally increased in subsequent scans (**Stejskal and Tanner, 1965**). Diffusion measurements with a static gradient system work in reverse, with the gradient strength constant and the diffusion encoding time, τ, incremented (**Carr and Purcell, 1954**). Both methods lead to a measured signal attenuation, an effect which can be summarized in a single variable, $b$ (**Stejskal and Tanner, 1965**). The diffusion MR signal from freely diffusing water with self-diffusion coefficient $D_0$ attenuates as $\exp(-bD_0)$ (**Woessner, 1961**), while water diffusing within restricted environments attenuates more slowly (**Wayne and Cotts, 1966**; **Robertson, 1966**; **Neuman, 1974**; **de Swiet and Sen, 1994**; **Hurlimann et al., 1995**). The diffusion MR signal of water in heterogeneous materials such as biological tissue would be expected to contain a multitude of components arising from water in different microenvironments, which restrict water diffusion to varying degrees (**Benjamini and Basser, 2017**).

Interpretation and modeling of the signal attenuation from diffusion measurements on neural tissue is an ongoing area of research (**Novikov et al., 2018**). Nonparametric data inversion techniques can model signal attenuation as arising from distributions of diffusion coefficients (**Pfeuffer et al., 1999**). This inversion assumes that the full attenuation is made up of a sum of multiexponenital attenuations, each with their own $D$ value. Distribution modeling can be a way to separate water components based on their apparent mobility. **Pfeuffer et al. (1999)**, along with **Ronen et al. (2006)** and **Benjamini and Basser (2019)**, suggest that the diffusion coefficient distribution can be used to investigate the microstructural properties of neural tissue.

Nuclear spins may also exchange along the diffusion coefficient distribution by moving between microenvironments, causing diffusion coefficients of components to shift and appear closer together on the distribution. Exchange can be measured from the change in apparent diffusion coefficients with encoding time (**Andrasko, 1976**; **Kärger et al., 1988**; **Waldeck et al., 1995**; **Pfeuffer et al., 1998**; **Thelwall et al., 2002**). Alternatively, MR can store the spin history from one encoding $b_1$ during a mixing time $t_m$ and recall it for a second encoding $b_2$ (**Cheng and Cory, 1999**) to measure exchange along the distribution (**Callaghan and Furó, 2004**). The standard diffusion measurement is one-dimensional (1-D) in that there is one encoding variable $b$ and one measured parameter $D$. By encoding the spins twice $(b_1, b_2)$, the diffusion exchange spectroscopy (DEXSY) sequence becomes 2-D (**Qiao et al., 2005**). 2-D DEXSY measures the relationship between spins' diffusion coefficients at two separate instances $(D_1, D_2)$ to show exchanging and non-exchanging components (**Bernin and Topgaard, 2013**; **Benjamini and Basser, 2017**).

The full DEXSY sampling of the 2-D $b_1 - b_2$ space (**Callaghan and Furó, 2004**; **Qiao et al., 2005**) is too time-consuming for scanning live specimen. Recent research shows that there is some redundancy in the data (**Bai et al., 2016a**; **Benjamini and Basser, 2016**; **Benjamini and Basser, 2018**; **Benjamini et al., 2017**) and alternative DEXSY-based approaches may measure exchange with fewer data points (**Aslund et al., 2009**; **Benjamini et al., 2017**; **Cai et al., 2018**). **Cai et al. (2018)** developed a rapid measurement of the exchange fraction, $f$, from just four points in the $b_1 - b_2$ space. **Aslund et al. (2009)** and **Lasič et al. (2011)** showed DEXSY-based methods measure the permeability of cell membranes to water.

Larger gradient strengths and gradient durations probe smaller structures (**Callaghan et al., 1991**). Hardware constraints cap the maximum strength of gradient coils of MRI systems at a few Tesla/m. With long gradient pulse durations and encoding times, diffusion MR microstructural resolution is predicted to be limited to structures larger than a few microns (**Nilsson et al., 2017**). However, when water within these structures exchanges on a timescale faster than the diffusion encoding time, the structure sizes appear inflated. In the extreme case of fast exchange, the structures are no longer visible as attenuation measures only the mean diffusivity (**Kärger et al., 1988**; **Yang et al., 2018**). Gradient pulses last at least a millisecond, which sets a lower limit for the encoding time (**Price, 1998**). Surprisingly, an experiment dating back to the origins of MR (**Hahn, 1950**; **Carr and Purcell, 1954**), performed in a strong static gradient field, can break this microstructural resolution limit (**Kimmich et al., 1991**). Low-cost, portable, bench-top, single-sided NMR devices with greater than 10 T/m static gradients (**Eidmann et al., 1996**) can probe sub-micron structures (**Carlton et al., 2000**) that ordinarily cannot be resolved from larger microscale structures using state-of-the-art pulsed gradient MR systems with lower maximum gradient strengths (**Potter et al.,**

*1996*). Displacement encoding within a static gradient field occurs by using radiofrequency (RF) pulses (*Hahn, 1950*; *Carr and Purcell, 1954*) to switch the 'effective gradient' (*Callaghan, 2011*), allowing for diffusion encoding times as short as 100 µs (*Stepišnik et al., 2018*). This permits resolution of sub-micron structures that can contain rapidly exchanging water pools (*Carlton et al., 2000*). The static gradient 1-D diffusion (*Rata et al., 2006*) and 2-D DEXSY (*Neudert et al., 2011*) experiments can then be used to probe cellular and sub-cellular components and water exchange between them.

In this paper, we adapt 1-D diffusion and 2-D diffusion exchange methods to perform measurements with a single-sided MR system having a strong static gradient, to investigate the cellular and sub-cellular structures in isolated neonatal mouse spinal cord. We develop a system to support both live and fixed spinal cords during NMR measurements such that data could be compared directly. We present both unprocessed raw signal data and processed diffusion coefficient distributions. Diffusion coefficient distributions show signal from water in various free and restricted environments. Diffusion signal attenuation can isolate signal from water restricted within structures smaller than a micron indicating that subcellular structural resolution is achieved. DEXSY measures the exchange of water between restricted and free environments on a timescale of 10 ms. Therefore, resolution of subcellular membrane structure requires encoding times less than 10 ms. Replacing the protons of water with deuterium ($D_2O$) decreased the signal from all components of the distribution equally, indicating that the majority of the signal is coming from water. Delipidation of membranes by the surfactant Triton X indicated that restriction was caused by lipid membranes and not proteins.

## Results

### System provides high sensitivity to motion and restricted motion within spinal cords

The single-sided magnet's field strength decreases rapidly with distance from the top surface, with a gradient of $g$ = 15.3 T/m. With diffusion measurements, $g$ provides a nominal resolution to displacements of water on the order of the dephasing length $l_g$ = 800 nm (see Appendix 1). Signal from water which diffuses an average distance $l_g$ significantly dephases and thus attenuates (subtracts from) the total measured signal. The diffusion encoding time τ is incremented to increase the average distance which water displaces $l_d = \sqrt{D_0\tau}$ relative to $l_g$ (where $D_0$ is the self-diffusion coefficient of freely diffusing water). One benefit of diffusion measurements with a strong static gradient is that free water signal is efficiently attenuated (the shortest τ for a given $g$), whereas signal from water which is restricted persists, allowing for the identification of restricted and free water at very short time and length scales. With $g$ = 15.3 T/m and $D_0 = 2.15 \times 10^{-9}\mathrm{m^2/s}$, freely diffusing water has significantly attenuated by τ>0.3 ms ($l_d$>$l_g$ = 800 nm) and the remaining signal is mostly made up of water for which diffusion is impeded on that time and length scale. At $b \times D_0 = 6$ (for which τ = 0.63 ms and free water moves on average $l_d = \sqrt{D_0\tau} = 1.16\ \mu m$), freely diffusing water signal has attenuated to exp (−6) = 0.0025 which is approximately the standard deviation of the noise. Signal at $b \times D_0 = 6$ (or the nearest data point) is used to define the restricted water fraction. Alternatively from diffusion coefficient distribution analysis, integrals of $P(D/D_0)$ on either side of $D/D_0 = 0.17$ are heuristically used as measures of the free and restricted water fraction.

Solenoid radiofrequency (RF) coils were specially built to the size of the spinal cords under study. In the solenoid coil, the spinal cord is oriented with its length perpendicular to the gradient such that the system measures diffusion of water perpendicular to the spinal cord. Artificial cerebro-spinal fluid (aCSF) bathes the spinal cord and RF coil and provides nutrients to live tissue.

The free diffusion coefficient is defined as the diffusion coefficient of aCSF at 25°C, $D_0 = 2.15 \times 10^{-9}$ m²/s, found by monoexponential fits (*Figure 1—figure supplement 1*). aCSF is well described by a single diffusion coefficient. Error residuals are random with standard deviation (SD) consistent with the noise of the system. Distributions of diffusion coefficients from data inversion are non-dimensionalized by $D_0$ such that the aCSF distribution should be a delta function at $D/D_0 = 1$. The inversion method smooths and broadens the aCSF distribution (*Figure 1c*) due to regularization which is needed to stabilize distribution estimates (*Provencher, 1982*).

Signal was acquired from a 400 µm slice through the spinal cord sample and aCSF bathing the sample. The contribution from the aCSF surrounding the sample needed to be quantified. *Figure 1*

shows distributions of diffusion coefficients for a fixed spinal cord placed within the RF coil and bathed in aCSF (a) and after removing the aCSF from the RF coil using a pipette and kimwipes, leaving only the sample and the fluid within the sample (b). These can be compared to the distribution from only aCSF filling the RF coil (c). Differences in the free diffusion component fraction indicates that aCSF accounts for only 5% of the signal in (a). The solenoid coil itself does an excellent job isolating signal from the spinal cord sample filling its interior.

## 1-D diffusion measures 25% of water to be restricted

Signal attenuation and diffusion coefficient distributions from 1-D diffusion measurements performed on a fixed spinal cord specimen are presented in *Figure 2*. Signal attenuation from measurements of pure aCSF is also shown for comparison. Signal is plotted as a function of the non-dimensionalized diffusion encoding variable $b \times D_0$. Exponential attenuation is expected for fluids diffusing freely. N.B. The largest $b$ ($\tau$ = 6.6 ms) corresponds to 3,000,000 s/mm$^2$, two to three orders

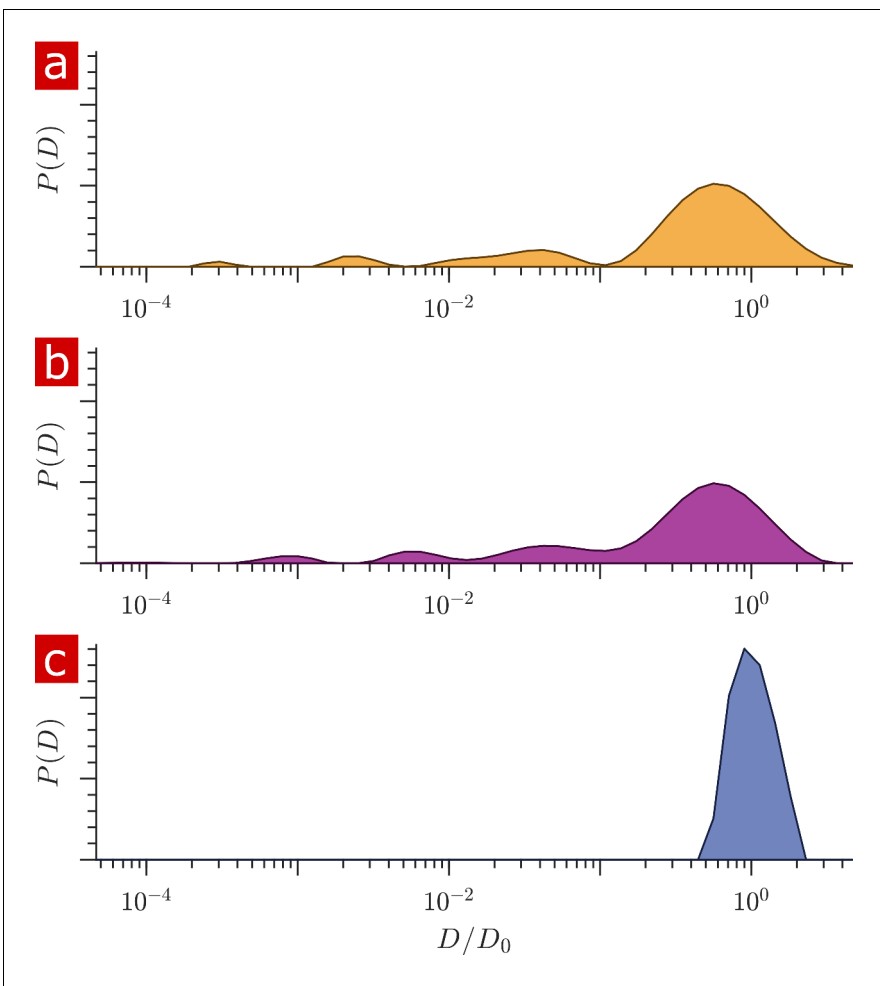

**Figure 1.** Diffusion coefficient distributions. (**a**) Distribution of diffusion coefficients from a fixed spinal cord bathed in aCSF. (**b**) Distribution from the same spinal cord after removing the aCSF bath. (**c**) Distribution from only aCSF filling the RF coil.

The online version of this article includes the following source data and figure supplement(s) for figure 1:

**Source data 1.** 1-D Diffusion signal attenuation (*I*) and *b* values for the measurements on fixed spinal cord bathed in aCSF (wet), the same spinal cord after removing aCSF (dry), and for aCSF (MATLAB structure array).

**Source data 2.** 1-D Diffusion signal attenuation (*I*) and *b* values for repeated measurements on aCSF (MATLAB structure array).

**Figure supplement 1.** Measurement of aCSF $D_0$ at 25˚C.

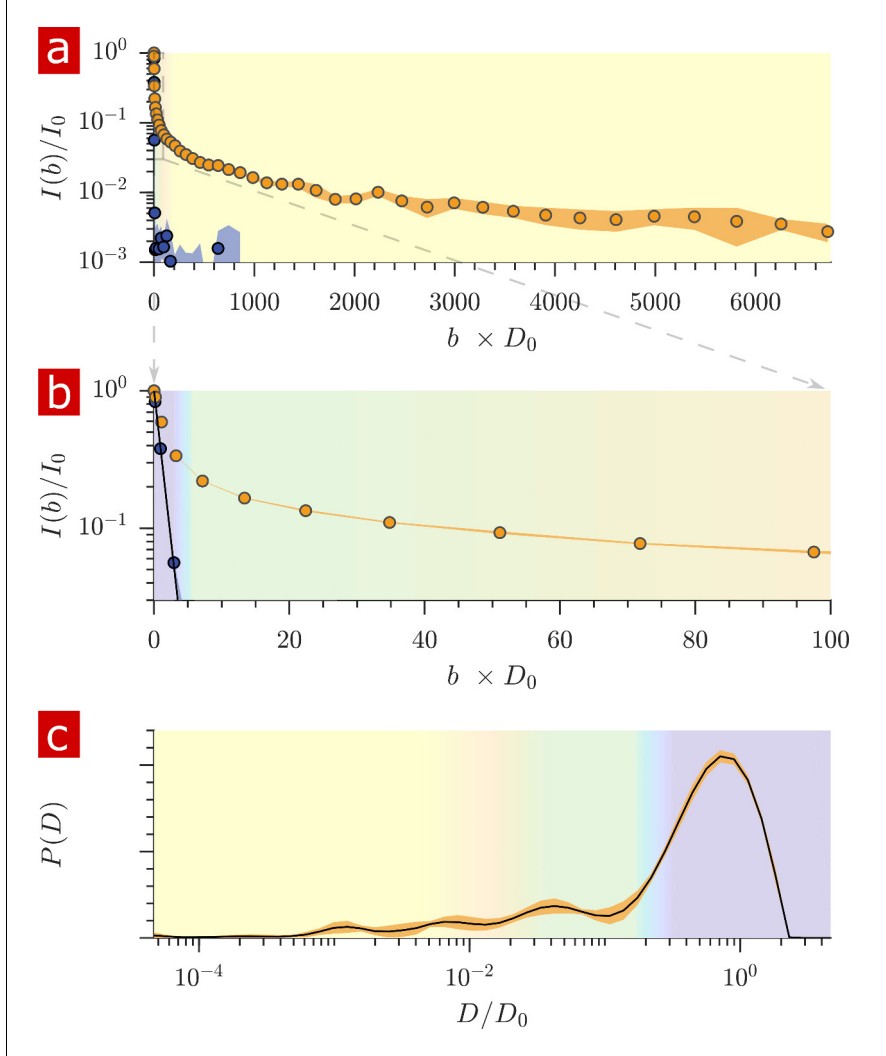

**Figure 2.** Diffusion measurements performed on a fixed spinal cord. (a) Mean (circles) and SD (shaded bands) of the signal intensity from five diffusion measurements, spaced 54 min apart, performed on a fixed spinal cord (orange) and three measurements performed on aCSF (purple) at 25° C. (b) Signal intensity from the zoomed in area shown in (a). (c) The distribution of diffusion coefficients resulting from inversion of the data. The purple, green, and yellow shading across plots signifies water which is free, less restricted, and more restricted, respectively. Models of signal attenuation (see Appendix 1) are used to define the cutoffs for each of these regimes based on when their signal components would attenuate to exp (−6). Values are inverted to define the color shading on the $D$ distribution. This inversion of colors is simply to guide the eye. The color gradient is meant to signify a continuous change between diffusion regimes rather than sharp boundaries.

The online version of this article includes the following source data and figure supplement(s) for figure 2:

**Source data 1.** 1-D Diffusion signal attenuation ($I$) and $b$ values for measurements repeated every 54 min on a fixed spinal cord (MATLAB structure array).

**Figure supplement 1.** Variability of diffusion data from measurements repeated on the same samples.

of magnitude larger than what is typically reached in conventional pulsed gradient diffusion MRI studies. aCSF signal is monoexponential and quickly attenuates to the background noise level. This background noise is Gaussian with mean = 0.001 and SD = 0.002 (*Figure 2—figure supplement 1c*). Spinal cord signal attenuation is multiexponential over the entire range of $b$ and does not fully attenuate, implying the presence of multiple highly restricted pools. System characteristics led to diffusion measurements with SNR > 500 such that signal could be observed at extremely high diffusion weightings (see Materials and methods). The signal intensity from 30 measurements performed on

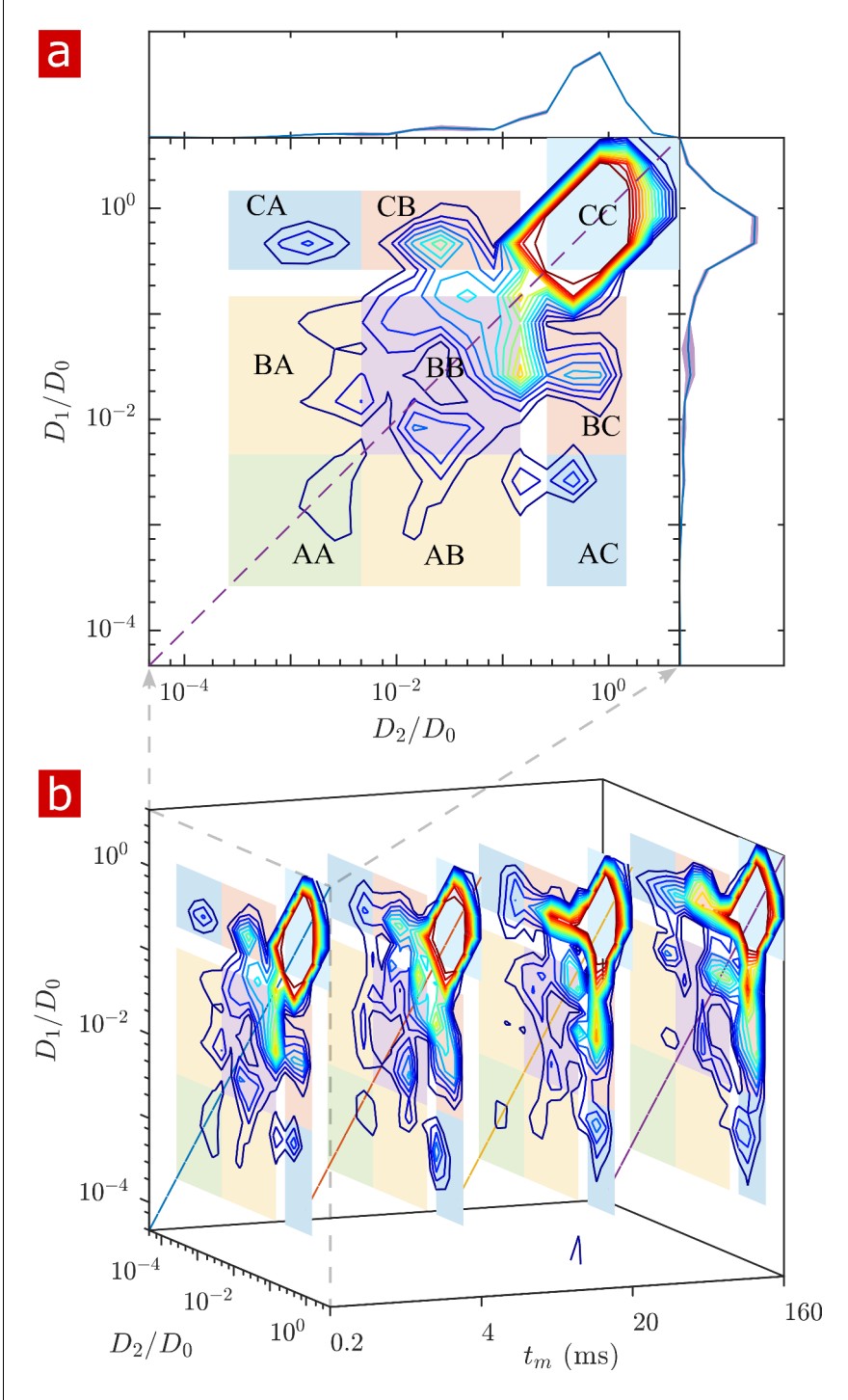

**Figure 3.** Full 2-D DEXSY diffusion exchange distribution for a fixed spinal cord. (a) Exchange distribution measured with mixing time $t_m = 0.2$ ms. Distributions show exchanging (off-diagonal) and non-exchanging (on-diagonal) components. These components are lumped into regions A, B, and C, shaded and labeled in a $3 \times 3$ grid for each exchange combination. The range of $P(D_1, D_2)$ is set to add detail to components A and B, but cuts off the top of the most mobile region CC. The marginal distributions $P(D_1)$ and $P(D_2)$ are presented on the sides, with mean (solid blue lines) and SD (shaded bands around lines) from three measurements. (b) A stacked view of distributions measured with mixing time $t_m = [0.2, 4, 20, 160]$ ms. With increasing $t_m$, the probability density builds up in regions of free and restricted water exchange $AC + CA$ and $BC + CB$ and decays for non-exchanging restricted water regions AA and BB.

*Figure 3 continued on next page*

*Figure 3 continued*

The online version of this article includes the following source data and figure supplement(s) for figure 3:

**Source data 1.** Full 2-D DEXSY datasets from measurements repeated three times at each mixing time on each of the fixed samples (MATLAB structure array).
**Figure supplement 1.** Fractions of exchanging and non-exchanging water.

the same fixed sample over the course of 30 hr varied similarly to the background noise (*Figure 2—figure supplement 1a and b*). Measurement variability on fixed samples is thus simply determined by the signal-to-noise ratio (SNR) and the system is amenable to long scans and signal averaging.

The distribution of apparent diffusion coefficients is shown in *Figure 2c*. The majority of the distribution is made up of free water. Humps extending to lower values of $D/D_0$ represent signal which is more and more restricted and on smaller length scales, as indicated by the color gradient.

Both the signal attenuation and the distributions show that the mobility of a large portion of water is restricted to some degree during the diffusion encoding time. The restricted fraction quantified from distribution analysis is (mean ± SD) 0.23 ± 0.006. Alternatively from raw signal, the restricted fraction is 0.22 ± 0.002. Taking into account a few percent of the free water component being from aCSF bathing the sample, roughly 25% of the water in the tissue is restricted on the 1 ms timescale.

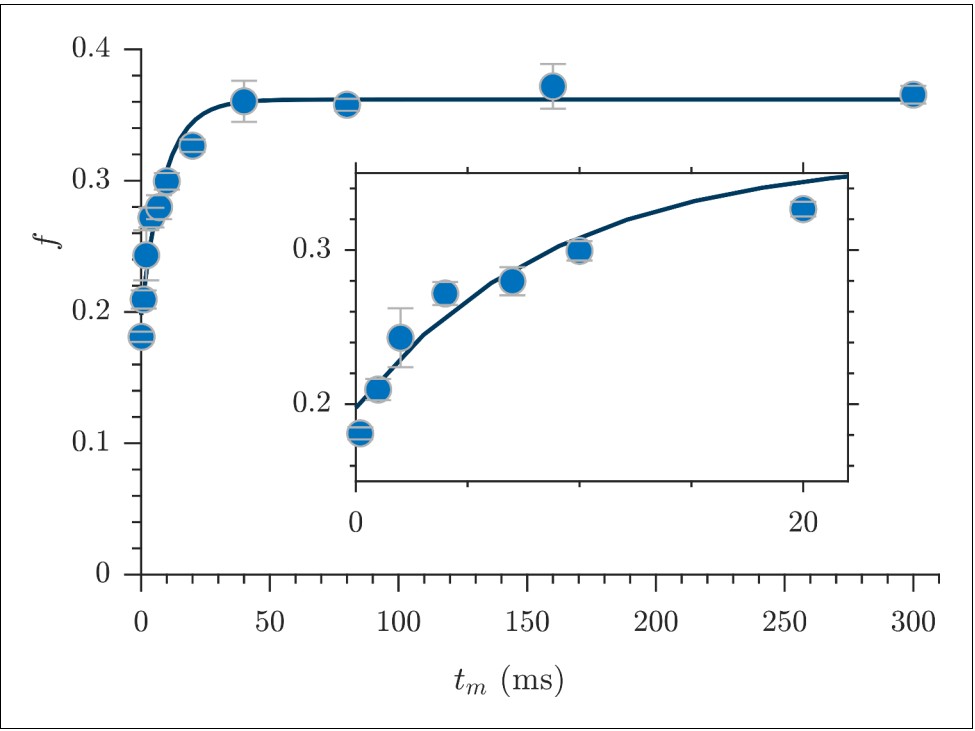

**Figure 4.** Rapid measurement of exchange fractions. A fit of the first-order rate model estimated an apparent exchange rate, AXR = 110 ± 30 s$^{-1}$ (mean ± SD from three measurements performed on one specimen at 25˚C). Inset shows a zoomed-in region of the initial rise in exchange.
The online version of this article includes the following source data and figure supplement(s) for figure 4:

**Source data 1.** Rapid exchange data for all fixed samples, including raw data from the four-point method (*I*) and the exchange fractions (*f*) (MATLAB structure array).
**Figure supplement 1.** Variability of apparent exchange rates (AXRs).

# Full DEXSY measures water exchanging between free and restricted environments 100 times per second

2-D DEXSY labels spins based on their local mobility at two instances which are separated by the mixing time variable $t_m$ (*Callaghan and Furó, 2004*). This permits the direct measurement of water movement from one environment (e.g. A) to another (B) as well as water moving in reverse (B to A) to fulfill mass conservation. In the case that water exchanges between environments on intermediate timescales (greater than the diffusion encoding time and less than the longitudinal relaxation time $T_1$ which causes spins to forget their encoding), the exchange increases and saturates as a function of $t_m$ (*Washburn and Callaghan, 2006*; *Cai et al., 2018*; *Williamson et al., 2019*). The classic analysis of DEXSY is as a joint 2-D probability distribution showing relationships between the apparent diffusion coefficients of water populations during the first encoding period, $D_1$ and the second encoding period $D_2$ (*Qiao et al., 2005*; *Benjamini et al., 2017*). Integrated probability density at a point or region $(D_1, D_2)$ indicates the probability of a spin being at $D_1$ during encoding one and $D_2$ during encoding 2. Non-exchanging water populations have $D_1 = D_2$, defining a diagonal line across the distribution, whereas exchanging water populations are located off the diagonal. A representative 2-D DEXSY distribution for a spinal cord is shown in *Figure 3a*. The distribution is divided into a 3 × 3 grid for the possible exchange pathways between components A ($2.6 \times 10^{-4} - 4.7 \times 10^{-2}$), B

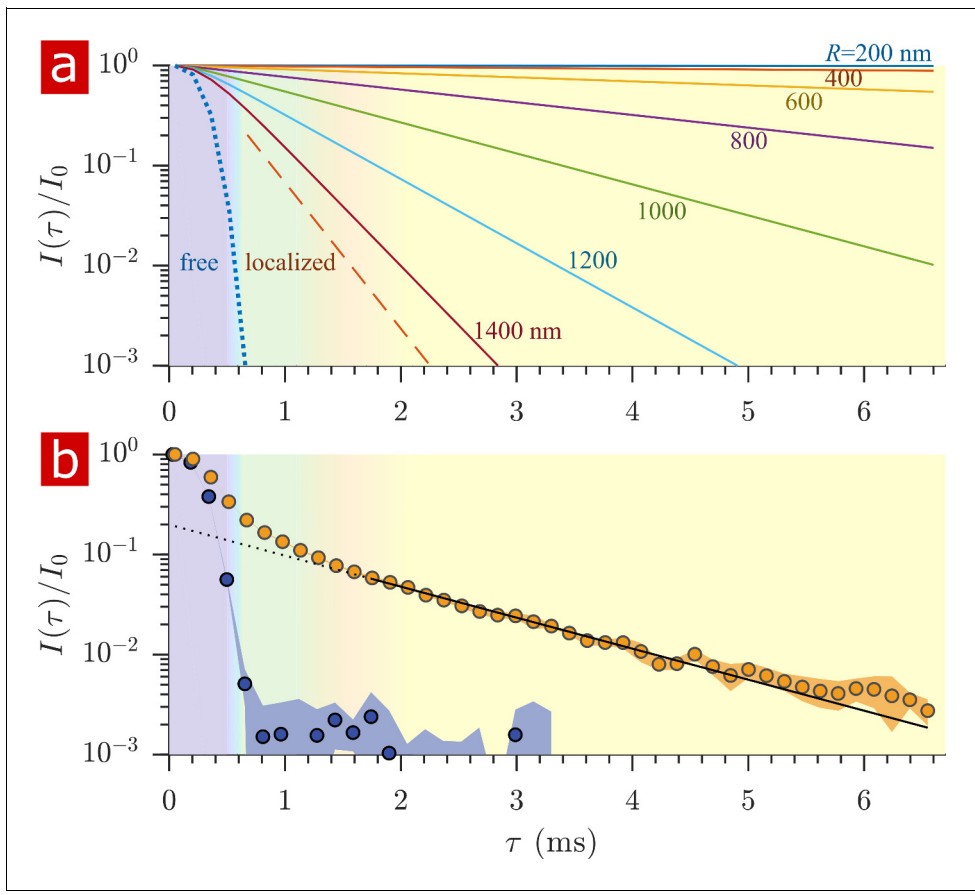

**Figure 5.** Sensitivity to membrane structure sizes from the diffusion signal attenuation. (a) Signal intensity is simulated for water restricted in spherical compartments of varying radius between R = 200 − 1400 nm (solid lines) (*Neuman, 1974*), for water localized near surfaces in larger restrictions (red dashed line) (*de Swiet and Sen, 1994*; *Hurlimann et al., 1995*), as well as water diffusing freely (purple dotted line) (*Woessner, 1961*). Signal is plotted as a function of the variable τ (rather than $b \sim \tau^3$). (b) Signal is re-plotted from *Figure 2*. Signal at τ ≥ 1.8 ms is fit with the model for water restricted in spherical compartments in the limit of long τ (solid black line) (*Neuman, 1974*) incorporating the AXR = 110 s$^{-1}$(*Carlton et al., 2000*), estimating a radius R = 900 nm. The dotted black line extrapolates back to $I/I_0 = 0.2$. (See Appendix 1 for model equations.) Color shading is similar to *Figure 2*.

$(4.7 \times 10^{-2} - 1.5 \times 10^{-1})$, and C $(2.6 \times 10^{-1} - 4.7 \times 10^{1} \ D/D_0)$, shown by the color coding and labels. This division was chosen in an attempt to separate the free water component (C) from the restricted water component, and to separate the restricted component into two groups (B and A) based on their apparent mobility. The integrated probability density from each region represents an exchange (off-diagonal) or non-exchange (on-diagonal) fraction. The distribution shows exchange between free water and restricted water. Additionally, there appears to be exchange between restricted components. Stacked plots at $t_m$= 0.2, 4, 20, and 160 ms (b) show the increase in probability for components exchanging with free water (regions $AC + CA$ and regions $BC + CB$) and a decrease in probability for the non-exchanging components (e.g. region $AA$ appears to decrease to near zero at the longest $t_m$).

The build-up of exchange fractions and decay of non-exchange fractions over $t_m$ are fit with a first order rate equation to obtain apparent exchange rates (AXRs) (*Figure 3—figure supplement 1*). Measurements on five different spinal cords show consistent exchange behavior. Restricted components exchange with free water with AXR $\approx$ 100 s$^{-1}$. $f_{AB+BA}$ does not increase with $t_m$, indicating that the DEXSY measurement is primarily sensitive to water exchanging between restricted and free environments and not between and among different restricted environments.

## Rapid exchange measurement agrees with full DEXSY

Full DEXSY measurements at four mixing times took 8 hr—too long to measure exchange in living tissue. Therefore, *Cai et al. (2018)* developed a method to rapidly measure exchange. The rapid measurement provides an apparent exchange fraction $f$, a diffusion-weighted average of exchange between all water pools. The full DEXSY can resolve multiple exchanging water pools and the exchange pathways between them (*Dortch et al., 2009*; *Van Landeghem et al., 2010*). Although the rapid measurement lacks the full DEXSY's resolution of multi-component exchange, it provides enhanced temporal resolution, both with respect to $t_m$ and experimental time, by sidestepping the need for 2-D data inversion (*Song et al., 2016*) and by acquiring the data much more rapidly. The protocol used here acquired the data at a rate of one exchange fraction (one $f(t_m)$) per minute.

Exchange fractions from the rapid measurement are presented in *Figure 4* for the same specimen as used for the full DEXSYs (*Figure 3* and *Figure 3—figure supplement 1*). The AXR from three repeat measurements was 110 ± 30 s$^{-1}$. This value is not different statistically from the results of the

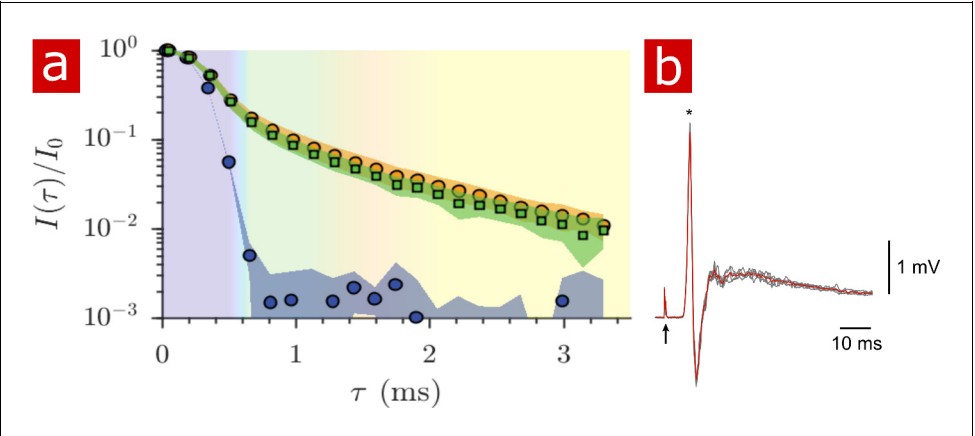

**Figure 6.** Diffusion in fixed vs. live. (a) Signal intensity from diffusion measurements performed at 25° C on live samples (n = 9) (green squares), fixed samples (n = 6) (orange circles) and aCSF (purple circles) plotted as a function of the variable τ. (b) Mono- and poly- synaptic reflexes were recorded from the L6 ventral root of live samples (n = 4) after NMR measurements. Stimulation was done on the homonymous dorsal root. The grey lines are five successive stimuli (30 s interval) and the superimposed red line is the average signal. The arrow indicates the stimulus artifact and the star the monosynaptic reflex.

The online version of this article includes the following source data and figure supplement(s) for figure 6:

**Source data 1.** 1-D diffusion data for all fixed and live samples (MATLAB structure array).
**Figure supplement 1.** Sample-to-sample variability of diffusion data.

full DEXSY measurement, validating the rapid measurement method. Additionally, the value from repeated measurements on five fixed samples was $110 \pm 20$ s$^{-1}$ at 25° C (mean $\pm$ SD taken across all $3 \times 5$ measurements), indicating high reproducibility between specimen. Variability of AXRs is presented in *Figure 4—figure supplement 1*.

## Restricted diffusion measures sub-micron structures

After free water has fully attenuated, restricted water signal attenuation is exponential with the diffusion encoding time τ and the size of the restriction (*Wayne and Cotts, 1966*; *Robertson, 1966*; *Neuman, 1974*; *de Swiet and Sen, 1994*; *Hurlimann et al., 1995*) (see model equations in Appendix 1). Attenuation models for water restricted in spheres of radius $R$ indicate that with $g$ = 15.3 T/m the diffusion experiment provides a 200–1400 nm window on restriction radii (*Figure 5a*). Signals from water in restrictions smaller than R = 200 nm do not attenuate significantly enough to differentiate. In restrictions larger than R = 1400 nm, signal from water far from surfaces attenuates as free water and signal from water near surfaces attenuates as localized water (restricted on one side but free on the other). The long-time behavior of the diffusion signals are analyzed to estimate a radius of restriction in *Figure 5b*. Exchange also causes attenuation which is exponential with τ (*Carlton et al., 2000*). The estimate accounts for attenuation due to exchange, utilizing the measured AXR. The estimated radius is R = 900 nm. This can be viewed as a volume-averaged restriction length, filtering out water in structures with R> 1400 nm.

## NMR recordings do not affect viability of spinal cord

The signal attenuation from diffusion measurements performed on live spinal cords (n = 9) is compared to that of fixed spinal cords (n = 6) and only aCSF in *Figure 6a*. The signal from live tissue attenuates slightly faster than signal from fixed tissue although not significantly as seen by the standard deviations. The mobility of water on the timescale of milliseconds is very similar in live and fixed specimen. Sample-to-sample variability of signal attenuation is presented in *Figure 6—figure supplement 1*.

After 2 hr of NMR measurements and 4 to 7 hr post-dissection, spinal cords (dissected on postnatal (P) day P2, P3, and P4) were assessed for viability by recording motoneuronal electrical activity after stimulation of a dorsal root. Mono- and polysynaptic reflexes were recorded in all preparations (n = 4), *Figure 6b*, indicating that neither the experimental setup nor the protocol compromised the neuronal excitability of the spinal cord.

## NMR measurements are primarily sensitive to water

To determine whether biomacromolecules were contributing to the signal observed in the spinal cords, rapid exchange and 1-D diffusion measurements were recorded in real-time as a fixed spinal cord was washed with aCSF made with 99.8% deuterium water (D$_2$O aCSF). (Results are presented in Appendix 3). After two successive washes, proton signal decreased to values similar to D$_2$O aCSF alone. All diffusion coefficient distribution components decreased similarly after D$_2$O washing. Components of the distribution which are not from water would still remain after removing H$_2$O. Therefore, all distribution components are primarily made up of water. This points to water rather than biomacromolecules accounting for the vast majority of the measured signal.

## Delipidation shows membranes to be the sole source of restriction

Triton X surfactant was used to remove lipid membranes from spinal cords in order to determine the effect of membranes on water restriction. The aCSF bathing the spinal cord was replaced with aCSF containing Triton X while rapid exchange and 1-D diffusion measurements were repeatedly performed (n = 2). *Figure 7* shows exchange fractions (a) and 1-D distributions from select time points (b). *Figure 7—video 1* shows the timelapse of diffusion coefficient distributions throughout the delipidation process.

After the addition of 1% Triton X at 1 hr, the exchange fractions decreased slowly and reached a plateau. Washing to 5% Triton X at 50 hr decreased the exchange fractions further until they again reached a plateau.

The diffusion coefficient distributions (*Figure 7b* and *Figure 7—video 1*) show that delipidation removes the barriers which restrict water mobility. Therefore, lipid membranes, not the remaining

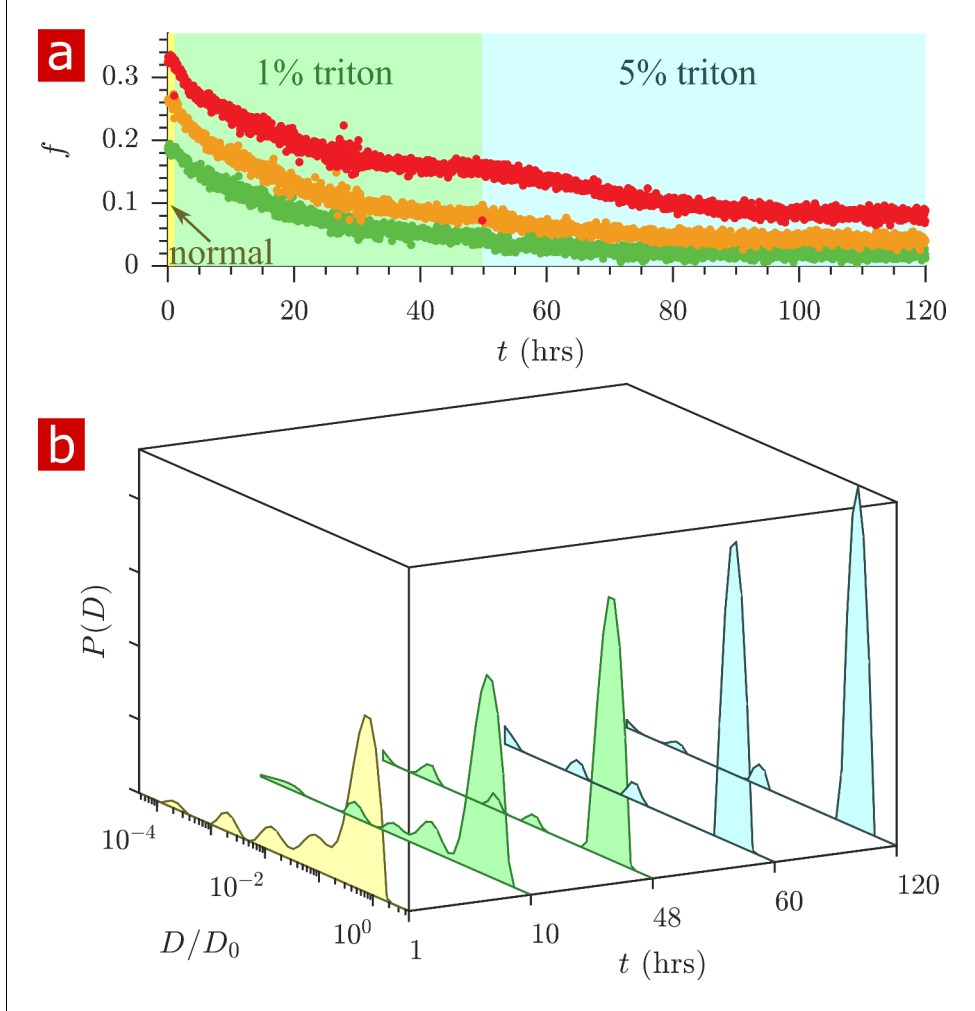

**Figure 7.** Timecourse study of Triton X delipidation. (a) Exchange fractions from rapid exchange measurements with $t_m = 0.2$ (green dots), 4 (orange dots), and 20 ms (red dots) measured throughout the timecourse, as the sample was washed to aCSF with 1% Triton X, and then 5% Triton X. (b) Representative diffusion coefficient distributions from 1-D diffusion measurements performed at different times before and after addition of Triton. The online version of this article includes the following video, source data, and figure supplement(s) for figure 7:

**Source data 1.** 1-D diffusion data for real-time delipidation of a fixed spinal cord (MATLAB structure array).
**Figure supplement 1.** Timecourse of restriction during Triton X delipidation.
**Figure supplement 2.** Diffusion coefficient distributions of 5% Triton X in aCSF.
**Figure 7—video 1.** Timelapse video of diffusion coefficient distributions during delipidation.
https://elifesciences.org/articles/51101#fig7video1

biomacromolecules, are the source of restriction of water diffusion. The fraction of restricted water (*Figure 7—figure supplement 1a*) decreases and plateaus similarly to the exchange fractions.

At 120 hr of delipidation, the diffusion coefficient distribution shows 6% restricted fraction, primarily from a component at $D/D_0 = 0.01$. This is signal from Triton X, which as a 5% solution imparts a 6% signal at $D/D_0 = 0.01$ (*Figure 7—figure supplement 2*).

Samples (n = 2) were also studied after full delipidation and washing away Triton X (*Figure 8*). The diffusion signal attenuation, (a) and (b), shows that 95% of the signal is monoexponential with $D/D_0 = 0.90$. The diffusion coefficient distribution (c) shows one major free diffusion peak which is not significantly different from the diffusion coefficient distribution of pure aCSF. Some small peaks which are not seen unless $P(D)$ is magnified lead to a 1% restricted component (also seen in the raw

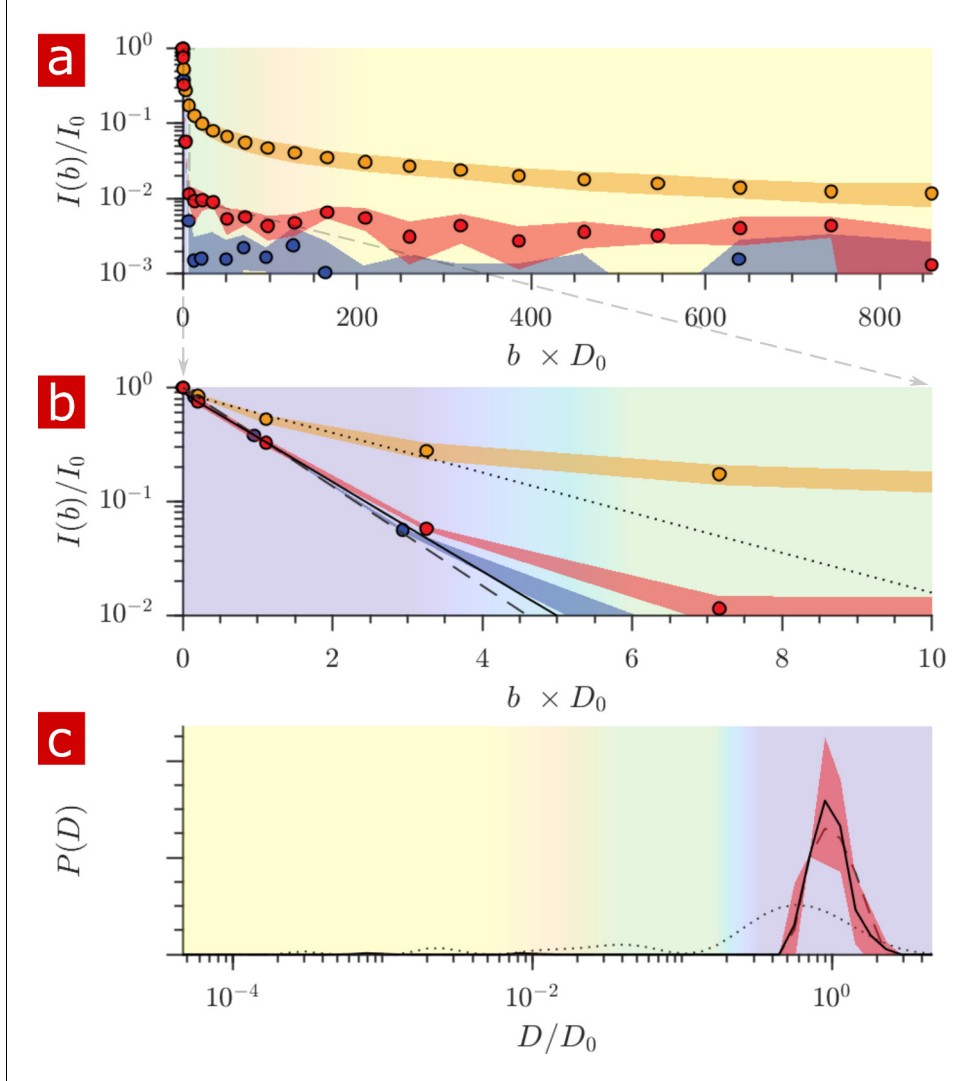

**Figure 8.** Diffusion measurement after delipidation. (a) Diffusion signal intensity from measurements on spinal cords performed at 25° C after delipidation (n = 2) with 10% Triton X and after washing the Triton away. The mean (circles) and SD (shaded bands) of the attenuation are plotted for the delipidated samples (red) alongside pure aCSF (purple) and fixed undelipidated spinal cords (n = 6) (orange). (b) The initial attenuation of signal. Monoexponential fits of the attenuation from points 2–4 yielded 2.15 ± 0.02, 1.94 ± 0.02 and 0.87 ± 0.14 ×$10^{-9}$ m$^2$/s for the aCSF, delipidated, and undelipidated spinal cords and are shown as the dashed, solid, and dotted lines, respectively. (c) Diffusion coefficient distribution of the delipidated spinal cords, for which the mean (solid line) and SD (shaded band around line) are not significantly different from the pure aCSF (dashed line). The distribution from a fixed, undelipidated spinal cord (dotted line) is also shown for comparison. The purple, green, and yellow shading across the plots signifies water which is free, localized, and restricted.

signal). This 1% component may be residual Triton X which remained after washing. Without membranes there is essentially only one diffusive environment throughout the whole sample.

## Discussion

We present NMR methods which use the diffusion of water to probe cellular and subcellular membrane structures on sub-millisecond and millisecond timescales. Much of the advance was possible because the strong static gradient overcomes (*Kimmich et al., 1991*) many hardware (*Price, 1998*) (e.g. slew rate, eddy currents, maximum gradient strength) and biological (e.g. peripheral nerve stimulation *Ham et al., 1997*) limitations of pulsed gradients. In 1996, *Köpf et al. (1996)* realized

this capability on biological tissue, performing diffusion measurements using a 50 T/m static gradient in the stray field of a superconducting 9.4 T magnet. Another stray field study by *Carlton et al. (2000)* used an 18 T/m static gradient to measure bacteria concentrations. Bacteria are roughly a micron in diameter, similar to subcellular structures in tissue. They noted the static gradient experiment provided more intracellular signal compared to pulsed gradient experiments on bacterial systems (*Potter et al., 1996*) due to less exchange during the shorter diffusion encoding time. We repurposed a low-field single-sided permanent magnet (*Eidmann et al., 1996*) which, due to its profiling capabilities (*Perlo et al., 2005*), has found a number of niche applications in materials science and engineering, biology and medicine, and cultural heritage (*Casanova et al., 2011*; *Danieli and Blümich, 2013*; *Rehorn and Blümich, 2018*). The large 15.3 T/m gradient allowed for the attenuation of tissue water signal below $I/I_0 = 0.01$ in a diffusion encoding time of 6.6 ms.

A large SNR and Gaussian zero mean noise was necessary to resolve slowly attenuating signals above the noise and to not confuse the signal with a baseline noise floor. In general, SNR > 100 is needed for diffusion coefficient distribution analysis (*Mitchell et al., 2012*) and as a rule-of-thumb this allows for resolution of populations comprising as little as 1% of the signal. Performance tuning led to very stable measurements and SNR > 500. These modifications included a 2000 echo CPMG readout, 25 µs echo time, a sample-specific solenoid RF coil, a wet/dry chamber without circulation, and noise reduction/isolation.

High SNR and system stability led to highly reproducible data (*Figure 2—figure supplement 1*). Based on the standard deviation of the normalized signal from repeat measurements, $\mathrm{SD}/I_0 = 1/\mathrm{SNR} = 0.002$, pathological or physiological events which cause slight variation of the diffusion-weighted signal may be detectable. This level of sensitivity is similar to in vivo diffusion MRI on state-of-the-art systems, for example *Nunes et al. (2019)* reported $\mathrm{SD}/I_0 = 0.003$ for diffusion functional MRI of the rat brain (one scan, 1.5 ×0.23 × 0.23 mm voxels) . The study of live ex vivo tissue removes the variability associated with in vivo studies such as blood flow and motion. The open design of the NMR experimental setup facilitates real-time measurements during perturbations to the sample.

The variability of 1-D diffusion signal was much larger across samples (*Figure 6—figure supplement 1*) than across measurements repeated on the same sample (*Figure 2—figure supplement 1*). The additional variability may be due to structural and size differences between samples. In contrast, rapidly measured AXRs showed similar variability across samples and across repeat measurements (*Figure 4—figure supplement 1*). This would indicate that AXR variability was primarily driven by SNR. Fluid far from restrictions does not exchange on the timescale of the measurement and does not impact the AXR. Therefore, unlike the diffusion signal, the AXR is insensitive to sample size differences and large-scale structural differences. In the exchange measurement, the heterogeneity of water mobility which is encoded on a 1 ms timescale fully exchanges and reaches a steady-state by 300 ms (*Figure 4*). The AXR is sensitive to average surface-to-volume and permeability characteristics which are quite local (within 10 s of microns) to the membranes. These characteristics appear similar between samples.

The solenoid RF coil permitted low-power, 2 µs RF pulses, high filling factor, and maximized signal from the spinal cord filling its interior relative to aCSF. Previous studies on live ex vivo neural tissue utilized MR imaging (*Buckley et al., 1999*; *Bui et al., 1999*; *Shepherd et al., 2002*; *Thelwall et al., 2002*; *Shepherd et al., 2009*; *Tirosh and Nevo, 2013*). However, their analysis was on regions of interest (ROIs) which encompassed the entire sample, indicating that no additional specificity was obtained from the imaging. Because most of the signal came from spinal cord tissue, imaging was not necessary. This let us achieve high SNR and sufficiently rapid measurements.

The wet/dry chamber kept the liquid environment still while the gas environment provided oxygenation. Previous diffusion MRI studies on live ex vivo neural tissue provided oxygen to the sample through perfused aCSF (*Buckley et al., 1999*; *Bui et al., 1999*; *Shepherd et al., 2002*; *Thelwall et al., 2002*; *Shepherd et al., 2009*; *Tirosh and Nevo, 2013*; *Bai et al., 2016b*). Media perfusion can cause convection artifacts in the diffusion measurement (*Fabich et al., 2018*). Researchers typically implemented start-stop diffusion MR protocols, with aCSF perfusion between MR measurements (*Shepherd et al., 2002*; *Shepherd et al., 2009*). However, a steady concentration of oxygen is preferable and better represents the in vivo environment. The wet/dry chamber provided a constant supply of oxygen to the tissue while avoiding convection artifacts, creating ideal conditions for diffusion measurements on live tissue.

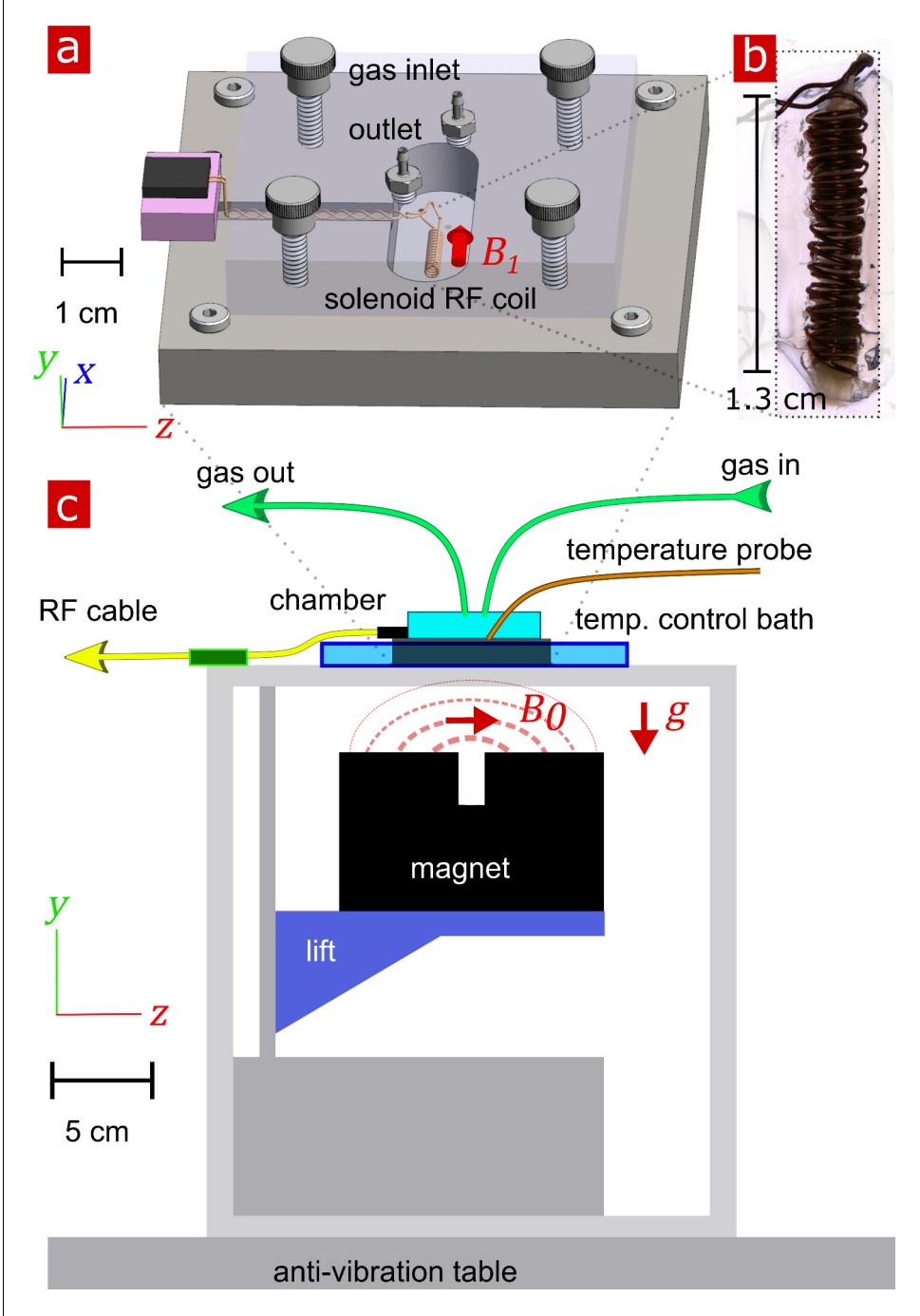

**Figure 9.** Experimental setup. (**a**) 3-D technical drawing of the test chamber. (**b**) Image of the solenoid RF coil containing a fixed, delipidated specimen. (**c**) Technical drawing of the experimental setup. The magnet is drawn in the 'service' position to show the field lines extending from one magnetic pole to the other. To perform measurements, the magnet would be raised such that the $B_0$ was correctly positioned relative to the sample. Vectors $B_1$, $g$ and $B_0$ point in the $x$, $y$, and $z$ directions respectively.

Neonatal mice were studied rather than adult mice because viability of ex vivo spinal cords is known to decrease with age (*Fulton, 1986*). Ex vivo spinal cords become more prone to hypoxia as they grow larger and oxygen in the tissue is consumed faster than it can diffuse to the center (*Wilson et al., 2003*). We directly demonstrate that spinal cords are alive after hours of NMR measurements by recording electrical responses from motoneurons.

Our novel experimental setup reveals signal from highly restricted pools. We determined that these pools are exclusively membrane-restricted water. Previous nerve tissue studies reported resolvable water mobility components spanning two orders of magnitude on the diffusion coefficient distribution (*Pfeuffer et al., 1999*; *Ronen et al., 2006*; *Benjamini and Basser, 2017*; *Benjamini and Basser, 2019*). Here, distributions showed components with diffusivities three orders of magnitude lower than free water. *Mailhiot et al. (2017)* reported similar values for other, larger, proton bearing molecules naturally occurring in biological tissue. However, by replacing the water in the tissue with deuterated water, we determined our methods to be sensitive primarily to protons on water and not protons on biomacromolecules.

Diffusion measurements with a strong static gradient separate free water from restricted water because freely diffusing water attenuates exponentially with $b$ whereas water which feels the boundaries of membranes attenuates exponentially with $b^{1/3}$ (*Grebenkov, 2018*) (which is proportional to $\tau$, as used for the abscissa in *Figure 5*) after the free water component has attenuated. The linearity of the attenuation at long $\tau$ (in *Figure 5*) provides an additional signature of restricted water.

Data inversion methods used to obtain 1- and 2-D distributions of diffusion coefficients assume that the data attenuates exponentially with $b$. This is only valid for a particular range of the signal attenuation, and a component that, for example attenuates exponentially with $b^{1/3}$ will appear to attenuate multiexponentialy with $b$ and thus result in a distributed component when inverted. Regularization is used to stabilize distributions in the presence of noise and has the effect of smoothing the distribution. The smoothing suppresses the artifactual 'pearling' of distributions into multiple peaks but broadens the peaks which should be sharp such as observed for aCSF (*Figure 1*) (*Williamson et al., 2016*). A conservative approach was taken in the regularization (discussed in the Materials and methods section) and when interpreting the distributions as showing free and restricted components. In particular, individual peaks in the restricted region are not interpreted as arising from individual compartment types or sizes. New development of nonparametric diffusion models may allow further interpretations.

After delipidation, the free water component accounts for 99% of the water, vs. roughly 75% of the water before delipidation (*Figure 8*). Signal attenuation is monoexponential with $\langle D \rangle / D_0 = 0.90$, vs. multiexponential with $\langle D \rangle / D_0 = 0.40$ before delipidation. Deviation from monoexponential, Gaussian signal attenuation is due to restriction by lipid membranes. From micron (*Beaulieu and Allen, 1994*; *Beaulieu, 2002*; *Leuze et al., 2017*) to sub-micron length scales, water restriction in tissue is due solely to membranes.

After removing restrictions, hindrances to water diffusion can be thought of as arising from volume obstruction by biomacromolecules such as proteins within the tissue. Obstruction models of water (solvent) self-diffusion incorporate only the volume fraction of biomacromolecules, $\theta$, as a free parameter and are generally adequate models in the limit of low ($<0.1$) volume fraction (*Masaro and Zhu, 1999*). Such models predict $0.03 < \theta < 0.07$ for $\langle D \rangle / D_0 = 0.90$. Neural tissue consists of 8% proteins (*Leuze et al., 2017*) and thus can account for the obstruction effects. *Leuze et al. (2017)* used another lipid clearing method to determine that lipids are the dominant source of MRI contrast. We can now additionally say that proteins act as simple obstructions, reducing water diffusion only slightly from $D_0$. This finding can be compared to previous reports of water diffusion in cytoplasm isolated from red blood cells, showing $\langle D \rangle / D_0 = 0.70$ (*Latour et al., 1994*). Although we used shorter diffusion encoding times, delipidated samples showed monoexponenital, Gaussian diffusion signal attenuation with $b$, indicating that microstructural information is averaged out during the encoding time and should show no additional encoding time dependence (*Novikov et al., 2014*). The decreased diffusivity from cytoplasm observed by *Latour et al. (1994)* could arise from the presence of organelles and membrane particles still present in the supernatant after lysing and centrifuging the red blood cells.

The methods were used to follow penetration of deuterated water into the tissue as well as the delipidation of the tissue via Triton X. The timescale of water penetration was $\approx$ 1 min, consistent with mass transport theory, but the timescale of Triton X penetration and delipidation was $\approx$ 1 day, longer than predicted ($\approx$ 3 hr) (see Appendix 4). An increased time is expected due to the reaction front which develops as Triton X delipidates, slowing its overall penetration.

No significant increase in exchange rate was observed during the delipidation timecourse. This is in contrast to reports of cationic and nonionic surfactant mixtures permeablizing the membranes of

yeast suspensions to water (*Lasič et al., 2011*). In the real-time delipidation of the spinal cord (*Figure 7*), a slowly progressing front of Triton X penetrating into the tissue removes all of the membrane structures as it passes. In the rapid exchange measurement, which is averaged over the saggital slice of the specimen, the most robust effect is the delipidation of compartments resulting in a reduction in exchange fractions.

Theoretical models indicate that the static gradient diffusion measurement provided a window on membrane structures smaller than roughly 1400 nm. The neonatal mouse spinal cord contains mostly gray matter (*Henry and Hohmann, 2012*; *Sengul et al., 2012*). Structures smaller than 1400 nm include cells and portions of cells with small radii such as axons, dendrites, glial processes, myelin, and a number of membranous organelles such as nuclei, mitochondria, endoplasmic reticula, and vesicles (*Kandel et al., 2013*), in addition to extracellular sheets and tunnels (*Kinney et al., 2013*).

The estimate of restriction length $R$ uses a theoretical model for the motional averaging regime and assumes that $R<l_g$ such that spins can diffuse across the restriction many times without significant dephasing. The localization regime, on the other hand, assumes $l_g<R$, such that water near surfaces moves and dephases slowly but can move to regions further away where it is free and dephases rapidly. $R \approx l_g$ would imply that the data falls into an intermediate regime between localization and motional averaging. Both regimes, and perhaps higher order terms to the models of attenuation in these regimes (*Grebenkov, 2007*; *Moutal et al., 2019*), may be playing a part in the attenuation. Therefore, the method used to estimate $R$ is simplistic. Nonetheless, the interplay between the attenuation regimes, heterogeneity of restriction sizes, and exchange make the modeling of diffusive motion of water in biological tissue a very challenging problem, and an important topic for future research.

The fixed direction of the gradient relative to the sample inhibits the study of anisotropy. Therefore, the restrictions imposed by long and slender cells and cell processes cannot be separated from restrictions of round or folded organelles. Further study is necessary to isolate the organelle contribution. Combined pulsed gradient and static gradient methods may serve beneficial for this purpose. Pulsed gradient methods can measure anisotropy through variation of the gradient directions (*Basser and Pierpaoli, 1996*; *Komlosh et al., 2007*). A combined study would be additionally beneficial by broadening the window of resolvable structure sizes (*Benjamini et al., 2014*; *Benjamini et al., 2016*; *Nilsson et al., 2017*).

Full 2-D DEXSY distributions showed water exchange between restricted and free pools, but could not resolve exchange between restricted water pools. The rapid measurement was designed to hone in on the exchange between restricted and free water pools and AXRs were consistent with results from the full DEXSYs. Results indicate that we have developed a non-invasive, sufficiently rapid method of measuring exchange across membranes in live tissue. The AXR $\approx 100$ s$^{-1}$ is significantly faster than intracellular–extracellular water exchange rates measured in neural tissue ($\approx 0.5 - 5$ s$^{-1}$) (*Quirk et al., 2003*; *Nilsson et al., 2013*; *Bai et al., 2018*; *Bai et al., 2019*; *Yang et al., 2018*). Such fast turnovers are not unheard of, for example red blood cells show similar ($\approx 100$ s$^{-1}$) rates (*Andrasko, 1976*; *Waldeck et al., 1995*; *Thelwall et al., 2002*) due to their high expression of aquaporin (*Kuchel and Benga, 2005*). Recently, *Veraart et al. (2018)* found that incorporating fast $30 - 100$ s$^{-1}$ exchange rates into compartmental models provided the best fit of human gray matter diffusion MRI data. They concluded that dendrites and unmyelinated axons which account for the majority of the neurites in gray matter have a greater permeability than myelinated axons which predominate white matter. In addition to membrane permeability, the other factor affecting exchange is the ratio of membrane surface to volume. This ratio increases with smaller structure sizes. Therefore, fast AXRs can be explained by the resolution of the system to membrane structures with high permeability and with large surface to volume ratios.

Signal attenuation from water which remains restricted during the timescale of diffusion encoding can provide sensitivity to structure sizes (*Assaf et al., 2008*). Studies indicate that pulsed gradient diffusion methods can measure the diameter of myelinated axons which are larger than a few microns (*Assaf and Cohen, 2000*). Exchange is on a long enough timescale to be neglected (*Nilsson et al., 2013*; *Novikov et al., 2014*) such that resolution can be treated as solely limited by gradient strength (*Nilsson et al., 2017*). The fast exchange rates measured in the current work indicate that extending structure size estimation methods to neural applications beyond myelinated axons requires that both gradient and diffusion encoding time be taken into consideration.

Significant exchange causes the size of structures to be overestimated. Since exchange rates increase with surface to volume ratio, the overestimation increases with decreasing structure size.

## Materials and methods

### Ethics statement for animal experimentation

All experiments were carried out in compliance with the National Institute of Neurological Disorders and Stroke Animal Care and Use Committee (Animal Protocol Number 1267–18).

### Test chamber and experimental conditions

The experimental test chamber (*Figure 9a*) was designed to support live spinal cord for hours without requiring oxygenated artificial cerebro-spinal fluid (aCSF) flow, thus avoiding flow-related measurement artifacts (*Fabich et al., 2018*). The gas-tight wet/dry chamber was fabricated at the NIH/NIMH mechanical workshop. The assembled chamber had two environments—a static liquid environment with aCSF and above it a gas environment with a slow flow of humidified 95% $O_2$ and 5% $CO_2$ gas. The sample temperature can be controlled in the range of $7 - 37°$ C. Sample temperature was monitored by a PicoM fiber optic sensor (Opsens Solutions Inc, Québec, Canada) and regulated by a shallow water bath surrounding the chamber. The bottom portion of the chamber was made of aluminum to provide good heat conduction to the media. See Appendix 5 for additional information.

In order to compare data between live and fixed tissue, oxygenated aCSF was used as the buffer solution for all experiments. A spinal cord was placed inside the solenoid RF coil within the chamber half-filled with aCSF previously bubbled with 95% $O_2$ and 5% $CO_2$. The chamber was sealed and connected to gas flow with humid 95% $O_2$ and 5% $CO_2$.

### Mouse spinal cord dissection, fixation, and delipidation

All experiments were performed on Swiss Webster wild type (Taconic Biosciences, Rensselaer, NY) between one day after birth to postnatal day 4. The mouse spinal cords were isolated and placed in a dissecting chamber perfused with cold Low-Calcium High Magnesium aCSF (concentrations in mM: 128.35 NaCl, 4 KCl, 0.5 $CaCl_2$ . $H_2O$, 6 $MgSO_4$ . $7H_2O$, 0.58 $NaH_2PO_4$ . $H_2O$, 21 $NaHCO_3$, 30 D-glucose) bubbled with 95% $O_2$ and 5% $CO_2$. To expose the spinal cords, a ventral laminectomy was performed, and they were subsequently isolated together with the ventral roots and ganglia. Spinal cords were roughly (anterior–posterior length $\times$lateral width $\times$ ventral–dorsal height) $15 \times 1 \times 1.5$ mm, increasing with days postnatal.

Prior to live spinal cord transportation, the cord was placed in a sealed 50 ml tube with 10 ml aCSF previously bubbled with 95% $O_2$ and 5% $CO_2$. The air in the tube was flushed with 95% $O_2$ and 5% $CO_2$.

For fixed experiments, at the end of dissection the cords were fixed in 4% paraformaldehyde overnight at $4°$ C. Fixative was then replaced with aCSF three times over the course of 2 days to remove any residual paraformaldehyde.

Triton X-100 (Sigma-Aldrich) nonionic surfactant was used to delipidate spinal cords. Samples (n = 2) were studied during delpidation by replacing the aCSF media with aCSF media containing a specified % of Triton X during NMR recording. Samples (n = 2) were also studied after delipidation with 10% Triton X in phosphate buffered saline (PBS) for 2 days, removal of Triton X by periodically replacing the PBS media for 2 more days, and equilibration in aCSF for a final day.

### NMR hardware

NMR measurements were performed at 13.79 MHz proton frequency with a Kea2 spectrometer (Magritek, Wellington, New Zealand). A PM-10 NMR MOUSE (Magritek, Aachen, Germany) permanent magnet (*Eidmann et al., 1996*) provided a $B_0$ magnetic field specially designed to be constant along an x–z (10 mm $\times$10 mm) plane parallel to the magnet's surface and to decrease rapidly and linearly in the y-direction perpendicular to the magnet's surface, providing a strong static magnetic field gradient (See *Figure 9c*) (*Perlo et al., 2005*). The NMR MOUSE was raised or lowered with a stepper motor with a step size of 50 μm in order to move $B_0 = 0.3239$ T, $\omega_0 = 13.79$ MHz, to the

precise depth within the sample (17 mm from the surface of the magnet). At this depth, the magnetic field gradient $g$ = 15.3 T/m, or 650 KHz/mm.

Double-wrapped (length inner diameter) 13 $\times$ 2 mm solenoid radiofrequency (RF) coils (*Figure 9b*) and an RF circuit were built in-house. The solenoid connected to the circuit board with detachable pin connectors. Tune and match used two trimmer capacitors with range 1–23 pF (NMAJ25HV, Knowles Voltronics). RF pulses were driven by a 100 W RF pulse amplifier (Tomco, Adelaide, Australia). See Appendix 5 and *Appendix 5—figure 1* for additional information and circuit design.

## NMR experimental methods

NMR measurements were performed in Prospa 3.22 (Magritek). For all measurements, repetition time (TR) = 2 s, 90°/180° pulse times=2 $\mu$s and amplitudes = −22 /- 16 dB, and 2000 CPMG echoes were acquired with 25 µs echo time. The acquisition time and dwell time were 4 and 0.5 µs, respectively, leading to roughly a 400 µm slice thickness. The lift was positioned such that the signal was at a maximum, thus providing a slice through the center of the solenoid. Signal was phased such that the component from the real channel was maximum and the mean of the imaginary channel component was zero. Measurements were performed at room temperature or else at a controlled temperature 25 $\pm$ 0.25° C when specified in figure captions.

Diffusion measurements were performed using the spin echo sequence (*Rata et al., 2006*) (*Appendix 5—figure 2a*). τ was incremented linearly from 0.05 to 6.55 ms in 43 data points (corresponding to $b$ values from 1.4 to 3,130,000 s/mm$^2$) or, for live and some fixed specimen, from 0.05 to 3.3 ms in 22 points. The Diffusion data associated with each figure is made available as source data (e.g. *Figure 6—source data 1*). See Appendix 5 for additional information.

The DEXSY sequence (*Appendix 5—figure 2b*) was written in-house and used eight phase cycle steps. For full 2-D DEXSY measurements (*Callaghan and Furó, 2004*), data points were acquired on a 21 $\times$ 21 grid by incrementing $\tau_1$ linearly from 0.200 to 3.3 ms in an inner loop and $\tau_2$ from 0.213 to 3.313 ms in an outer loop. For the rapid exchange measurement (*Cai et al., 2018*), points were acquired as a function of $b_s$ and $b_d$ by varying $\tau_1$ and $\tau_2$ accordingly. The standard 4-point acquisition used one point at $b_s$ = 200, $b_d$ = 20 s/mm$^2$, and three points along $b_s$ = 4500 s/mm$^2$ with $b_d$ = −4300, −150, and $b_d$ = 4300. Unless otherwise specified, the $t_m$ list was [0.2, 4, 20, 160] ms for full DEXSYs and [0.2, 1, 2, 4, 7, 10, 20, 40, 80, 160, 300] ms for the rapid exchange measurement. DEXSY and rapid exchange data associated with each figure are made available as source data. The Prospa (V 3.22) DEXSY pulse program and macros for acquiring full DEXSY and rapid exchange data, and MATLAB (2019b) routines for compiling the data and fitting exchange rates are made available in a Supporting Zip Document. See Appendix 5 and *Appendix 5—table 1* for sequence details and phase cycles.

Standard CPMG $T_2$ (10 s TR, 8000 echoes) and saturation recovery $T_1$ (1 s TR, 21 recovery points logarithmically spaced to 10 s) measurements were performed, with all other parameters consistent with diffusion and exchange measurements. The data was fit with a monoexponential. Representative data and fits are shown inee *Appendix 6—figure 1* and *Appendix 6—figure 2*. The $T_2$/$T_1$ values were 275 $\pm$ 5/1870 $\pm$ 10 ms for aCSF (three measurements), 55 $\pm$ 13/972 $\pm$ 53 ms for fixed spinal cords (n = 10/4), and 176 $\pm$ 35/1030ms for fixed spinal cords after delipidation (n = 3/1).

## NMR data analysis

1-D distributions were fit using $\ell_2$ regularization (*Provencher, 1982*) and singular value decomposition (*Venkataramanan et al., 2002*; *Godefroy and Callaghan, 2003*), with 50 grid points logarithmically spaced from $10^{-13}$ to $10^{-8}$, and the regularization parameter chosen using the generalized cross validation (GCV) method (*Golub et al., 1979*). 2-D distributions were fit with an algorithm which uses $\ell_2$ regularization and singular value decomposition (*Venkataramanan et al., 2002*; *Godefroy and Callaghan, 2003*), with 21 $\times$ 21 grid points logarithmically spaced from $10^{-13}$ to $10^{-8}$ and the regularization parameter chosen by the L-curve method (*Mitchell et al., 2012*) and held constant for all experiments. Exchange fractions were calculated from the rapid exchange measurement using $D_e = 10^{-9}$ and $D_i = 10^{-11}$ m$^2$/s. AXRs from both full DEXSYs and the rapid exchange measurement were estimated from fits of a first-order rate model (*Washburn and Callaghan, 2006*; *Benjamini et al., 2017*; *Cai et al., 2018*), incorporating a non-zero initial condition to account for

exchange during encoding (*Williamson et al., 2019*). All analyses were performed using MATLAB (MathWorks). See Appendix 5 for additional information.

## System characteristics led to high SNR diffusion measurements

Although SNR is highly dependent on the magnetic field strength, the decrease in SNR at low field is boosted by refocussing the signal 2000 times in a CPMG train for each data point (*Rata et al., 2006*). Moreover, the solenoid RF coil maximized the sample filling factor, increasing SNR roughly 10-fold from previous flat RF designs (*Bai et al., 2015*). RF pulses used little power, permitting short $2~\mu s$ RF pulse durations and producing negligible heat. The coil design allowed for short echo times which reduced relaxation during acquisition in the CPMG train, again boosting SNR. Significant attention was given to shielding and grounding the equipment to minimize noise pickup. All together, 1-D diffusion measurements obtained SNR > 500. See *Figure 2—figure supplement 1* and Appendix 2 for additional information on noise and SNR.

## Electrophysiological recording

Electrical activity from motoneurons was recorded with suction electrodes into which individual ventral roots (L6 or T10) were drawn after NMR measurements (n = 4). The recorded signals were filtered (between 0.1 and 3 kHz) and amplified (gain: 1000), digitized at 10 kHz (Digidata 1500 B) and stored digitally on a computer. Episodes of data were analyzed off-line using MATLAB. To elicit monosynaptic responses in motoneurons, the homonymous dorsal roots were stimulated with a single electrical pulse (250 $\mu s$ duration) repeated 5 times at 30 s intervals. The threshold for a given spinal root was defined as the lowest current intensity at which that root had to be stimulated to elicit a monosynaptic response in 5/5 attempts. Recordings were obtained at 5 $\times$ threshold.

## Statistical analysis and reproducibility

The Results section presents data from multiple measurements repeated on individual specimen as well as measurements performed on groups of samples with different treatments (n = x). Each sample/specimen corresponds to one mouse spinal cord. The number of samples for each treatment group and (measurement type) were 9/6/5/1/2/2 for live (diffusion)/fixed (diffusion)/fixed (full DEXSY and rapid exchange)/fixed $D_2O$ wash (diffusion and rapid exchange)/fixed delipidation timecourse (diffusion and rapid exchange)/fixed delipidated 10% Triton (diffusion). Means and standard deviations (SD) are presented to quantify repeat measurement and sample-to-sample reproducibility of the results.

# Acknowledgements

We thank Randall Pursley, Danny Trang, and Marcial Garmendia-Cedillos for electrical engineering help, Sarah Avram for Triton X delipidation protocol advice, Alexandru Avram and Miki Komlosh for general suggestions, Uzi Eliav for phase cycling expertise, Elizabeth Murphy for resource provisions, and Petrik Galvosas for the 2D inversion code. Special thanks to Velencia Witherspoon for suggesting the solenoid coil and the $D_2O$ wash. NHW thanks attendees of the 2019 Tissue Microstructure Imaging Conference GRC for their input. NHW, DB and PJB were supported by the IRP of the NICHD, NIH. DB was also supported by the Center for Neuroscience and Regenerative Medicine, Henry Jackson Foundation, Bethesda, MD. HM, MF, MJO were funded by NINDS. DI was funded by NIMH.

# Additional information

## Funding

| Funder | Grant reference number | Author |
| --- | --- | --- |
| Eunice Kennedy Shriver National Institute of Child Health and Human Development | Intramural Research Program (IRP) | Nathan H Williamson<br>Rea Ravin<br>Dan Benjamini<br>Teddy X Cai<br>Ruiliang Bai<br>Peter J Basser |

| National Institute of Neurological Disorders and Stroke | Intramural Research Program (IRP) | Hellmut Merkle<br>Melanie Falgairolle<br>Michael James O'Donovan<br>Dvir Blivis<br>Dave Ide |
| National Institute of Mental Health | Intramural Research Program (IRP) | Dave Ide |
| National Heart, Lung, and Blood Institute | IRP | Nima S Ghorashi |

The funders had no role in study design, data collection and interpretation, or the decision to submit the work for publication.

### Author contributions
Nathan H Williamson, Conceptualization, Data curation, Formal analysis, Investigation, Visualization, Methodology; Rea Ravin, Conceptualization, Data curation, Formal analysis, Supervision, Investigation, Methodology; Dan Benjamini, Conceptualization, Software, Formal analysis, Visualization, Methodology; Hellmut Merkle, Dave Ide, Resources, Methodology; Melanie Falgairolle, Formal analysis, Investigation, Visualization, Methodology; Michael James O'Donovan, Conceptualization, Resources, Supervision, Funding acquisition; Dvir Blivis, Teddy X Cai, Ruiliang Bai, Conceptualization, Methodology; Nima S Ghorashi, Validation, Methodology; Peter J Basser, Conceptualization, Supervision, Funding acquisition, Investigation, Project administration, Writing - review and editing

### Author ORCIDs
Nathan H Williamson (iD) https://orcid.org/0000-0003-0221-9121
Melanie Falgairolle (iD) http://orcid.org/0000-0001-5243-4714
Michael James O'Donovan (iD) http://orcid.org/0000-0003-2487-7547
Dvir Blivis (iD) http://orcid.org/0000-0001-6203-7325
Peter J Basser (iD) https://orcid.org/0000-0003-4795-6088

### Ethics
Animal experimentation: All experiments were carried out in compliance with the National Institute of Neurological Disorders and Stroke Animal Care and Use Committee (Animal Protocol Number 1267-18).

### Decision letter and Author response
Decision letter https://doi.org/10.7554/eLife.51101.sa1
Author response https://doi.org/10.7554/eLife.51101.sa2

## Additional files

### Supplementary files
• Transparent reporting form

### Data availability
Source data for all Figures 1-9 in the manuscript have been provided as supporting files.

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

## Appendix 1

### Diffusion signal attenuation models

Attenuation of the MR signal in a spin echo diffusion experiment under a static magnetic field gradient can exhibit three diffusion regimes corresponding to three different characteristic length scales: the restriction length $l_s$, the diffusion length, $l_d = \sqrt{D_0 \tau}$, and the dephasing length, $l_g = (D_0/\gamma g)^{1/3}$ where $D_0 = 2.15$ m$^2$/s is the self-diffusion coefficient of water in artificial cerebro-spinal fluid (aCSF) at 25° C, $\tau$ is the time between the first 90° and the 180° radio frequency refocussing pulses of the spin echo sequence or 1/2 the echo time, and $\gamma$ is the gyromagnetic ratio (*Hurlimann et al., 1995*). The diffusion length is the average distance that water diffuses during the time $\tau$. In the spin echo diffusion measurements, $\tau$ was linearly increased from 0.05 to 3.3 or to 6.6 ms, corresponding to $l_D = 0.33$, 2.7, and 3.7 $\mu$m respectively. The dephasing length is the distance that diffusing spins dephase by $2\pi$ radians in a spin echo measurement with a static gradient. For the 15.3 T/m gradient used here, $l_g = 800$ nm. The shortest of these three length scales determines the regime that applies to the diffusing spins, and thus dictates the asymptotic behavior of the spin echo decay. What follows is a basic interpretation of the attenuation regimes, as taken from *Hurlimann et al. (1995)*. A complete discussion of the regimes, higher order terms to the attenuation, and a historical account of the field was provided by *Grebenkov (2007)*.

### Free diffusion regime

The free diffusion regime occurs when $l_D$ is the shortest characteristic length scale. This regime was first described by *Hahn (1950)* and then the diffusion coefficient of water was measured by *Carr and Purcell (1954)*, both using static magnetic field gradients. In this regime, *Woessner (1961)* showed the signal decays by

$$
\begin{aligned}
I(\tau)/I_0 &= \exp\left(-\frac{2}{3}D_0 \gamma^2 g^2 \tau^3\right) \\
&= \exp\left(-\frac{2}{3}\left(\frac{l_D}{l_g}\right)^6\right) \\
&= \exp\left(-bD_0\right).
\end{aligned}
\tag{1}
$$

Water diffusion is often modeled as Gaussian with an effective or apparent diffusion coefficient $D$, rather than $D_0$, and in the limit of low attenuation $D = \langle D \rangle$. 'Apparent' implies that the measured diffusion coefficient will depend on the experimental parameters (*Tanner, 1978*). This is particularly true when using *Equation 1* to model signal which includes water in other regimes. The use of $b$ coefficient or factor comes from diffusion MRI literature (*Le Bihan et al., 1986*).

### Localization regime

The localization regime occurs when $l_g$ is the shortest characteristic length scale. In this regime, signal near the restrictive surfaces will dephase more slowly than signal farther away. While the entire decay curve can be quite complicated (*Moutal et al., 2019*), in the asymptotic long-time ($\tau$) limit the signal was experimentally characterized by *Hurlimann et al. (1995)* and theoretically modeled by *Stoller et al. (1991)*, and *de Swiet and Sen (1994)*, and shown to attenuate as

$$
\begin{aligned}
I(\tau)/I_0 &= c\frac{D_0^{1/3}}{\gamma^{1/3}g^{1/3}l_s}\exp\left(-a_1 D_0^{1/3}\gamma^{2/3}g^{2/3}\tau\right) \\
&= c\frac{l_g}{l_s}\exp\left(-a_1\left(\frac{l_D}{l_g}\right)^2\right),
\end{aligned}
\tag{2}
$$

where $a_1 = 1.0188$ and is, importantly, independent of the confining geometry (*Moutal et al.,*

**2019**). The prefactor $c$ varies depending on the geometry and equals 5.8841 for water restricted between parallel plates (**de Swiet and Sen, 1994**). Note that $l_s$ affects the fraction of signal present in the asymptotic limit but does not affect the decay. Higher order terms, shown in **Moutal et al. (2019)**, do depend on geometry of the confining surface, in particular the curvature, permeability, and surface relaxivity. By varying τ under a static gradient, we see that the signal attenuates exponentially with τ, $(l_D/l_g)^2$ or $(bD_0)^{1/3}$.

## Motional averaging regime

The motional averaging regime occurs when $l_s$ is the shortest characteristic length scale. Signal attenuates very slowly and water can diffuse across the restricted volume many times before dephasing appreciably. Signal decay in the motional averaging regime was first experimentally measured by **Wayne and Cotts (1966)** and subsequently modeled by **Robertson (1966)**. **Neuman (1974)** derived the signal attenuation models for water restricted between parallel plates and within cylinders oriented perpendicular to $g$, and within spheres. We focus on the model for spheres of radius $R$ for which the signal attenuates by

$$I(\tau)/I_0 = \exp\left(-\frac{2\gamma^2 g^2}{D_0}\sum_{m=1}^{\infty}\frac{\alpha_m^{-4}}{\alpha_m^2 R^2 - 2}\left(2\tau - \frac{3 - 4\exp(-\alpha_m^2 D_0 \tau) + \exp(-\alpha_m^2 D_0 2\tau)}{\alpha_m^2 D_0}\right)\right) \tag{3}$$

where $\alpha_m$ is the $m$th root of

$$\alpha_m R J'_{3/2}(\alpha_m R) - \frac{1}{2}J_{3/2}(\alpha_m R) = 0 \tag{4}$$

for which the first 5 roots are $\alpha_m R =$ 2.0815, 5.940, 9.206, 12.405, and 15.579 (**Carlton et al., 2000**). In the limit of long τ relative to the timescale to diffuse across the restriction, **Equation 3** becomes

$$I(\tau)/I_0 = \exp\left(-\frac{8}{175}\frac{R^4\gamma^2 g^2}{D_0}\left(2\tau - \frac{581}{840}\frac{R^2}{D_0}\right)\right)$$

$$\approx \exp\left(-\frac{4}{175}\left(\frac{l_D}{l_g}\right)^2\left(\frac{l_s}{l_g}\right)^4\right) \tag{5}$$

where the final approximation drops the $(581R^2)/(840D_0)$ as insignificant. In the long-time limit, decay models for other geometries vary from **Equation 5** by a scaling within the exponential, for example rather than 8/175 for spheres, the scaling factor is 1/120 for parallel plates, and 7/296 for cylinders (**Neuman, 1974**).

As in the long-time limit of the localization regime, decay of signal in the motional averaging regime is exponential with τ, $(l_D/l_g)^2$ or $(bD_0)^{1/3}$. Exchange also occurs on the timescale of τ and **Carlton et al. (2000)** incorporated this into the decay model by multiplying **Equation 5** by exp (−2τ AXR) where AXR is the apparent exchange rate.

The effect of motional averaging can be reached in the extreme case of when δ approaches Δ in pulsed gradient measurements, which researchers have commented leads to restrictions appearing smaller than they actually are **Codd and Callaghan (1999)**; **Ryland and Callaghan (2003)**; **Malmborg et al. (2004)**. Given the gradient strength limitations on human MRI scanners, the clinical translation of advanced diffusion MRI methods requires the use of gradient pulses with maximum amplitude for efficient diffusion encoding **Avram et al. (2013)**. Consequently, clinical implementations of many advanced diffusion MRI preparations can be adjusted to effectively resemble/be equivalent to experiments in a static gradient field.

Decay models for the various regimes are compared in **Figure 5** of the main manuscript.

## Appendix 2

### Noise and SNR

The noise is quantified by the standard deviation of measured output from the real channel from the standard diffusion measurement protocol defined in the Materials and methods section (2 s TR, 2000 echoes, four scans, etc.), normalized by the average signal from diffusion scans with the the weakest diffusion weighting ($\tau = 0.05$ ms) ($\mathrm{SD}(I(\tau))/\mathrm{mean}(I_0)$). Signal-to-noise is defined as the inverse of the noise statistic ($\mathrm{SNR} = \mathrm{mean}(I_0)/\mathrm{SD}(I(\tau))$). *Figure 2—figure supplement 1* explores noise and SNR from the standpoint of repeatability of measurements performed on a specimen over the course of 2 days, the standard deviation of noise from measurements on aCSF after aCSF signal has attenuated, and system noise on an empty coil in a dry chamber. The SD of repeat measurements on a spinal cord appears similar to the SD of noise from measurements on aCSF. The signal-to-noise ratio of the system based on diffusion measurements on aCSF is $\mathrm{SNR} = I_0/SD = 500$. The theoretical maximum SNR = 1000 based on the SD of the noise from a dry empty chamber with the circuit tuned on resonance and $I_0$ defined by the aCSF experiment. Anderson–Darling tests indicate that the histograms from repeat measurements in *Figure 2—figure supplement 1* are not dissimilar from Gaussian distribution (accepting the null hypothesis with $P = 0.18 > 0.05$ for the spinal cord data in (b), $P = 0.08 > 0.05$ for the aCSF data in (c), and $P = 0.54 > 0.05$ for the empty chamber data in (d). In the case of aCSF, the noise has a non-zero mean (0.0011) indicating a 0.1% noise floor, however the empty dry chamber shows zero mean noise. The noise floor from aCSF may be signal measured from an immobile component (e.g. glue or a coating on the RF coil).

## Appendix 3

### Deuterated water wash

In order to determine the significance of signal from proton-bearing molecules other than water, the signal from water was diminished by washing from aCSF made with $H_2O$ to aCSF made with 99.8% deuterium $D_2O$ in two steps, and then back to $H_2O$ aCSF. For each washing step, the volume of aCSF was replaced twice. *Appendix 3—figure 1* shows results of proton density (a), exchange fractions at various mixing times (b) and diffusion coefficient distributions presented as signal fractions (c)) from rapid exchange measurements and 1-D diffusion measurements performed in real-time during the washes. Proton signal decreased to similar values to $D_2O$ aCSF alone, indicating that molecules other than water do not add a significant signal component. All diffusion coefficient distribution peaks decreased after washing with $D_2O$ indicating that the distribution represents water mobility, with no single component being solely from non-water molecules.

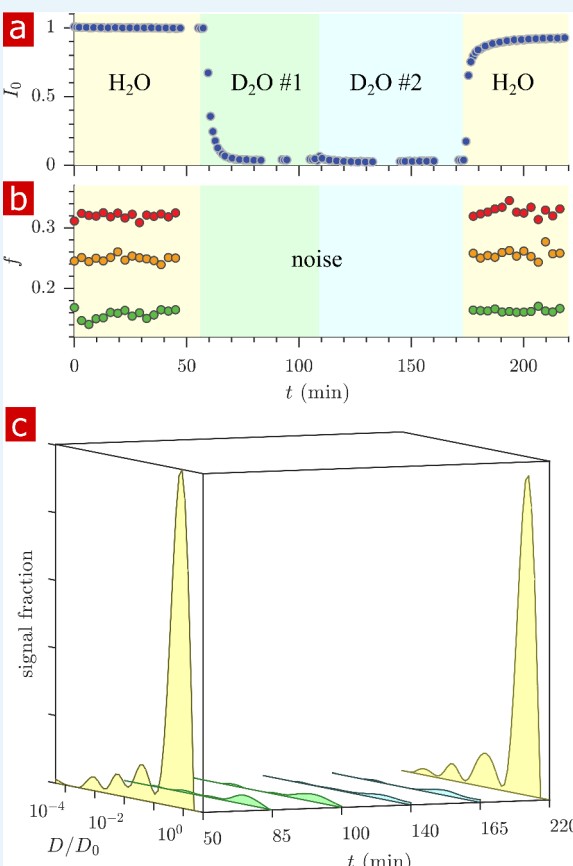

**Appendix 3—figure 1.** Timecourse study of $D_2O$ wash. The sample was washed from aCSF to aCSF made with deuterated water in two steps and back to aCSF as shown by the yellow, green, blue, and yellow pastel color shadings for $H_2O$, $D_2O$ #1, $D_2O$ #2, and $H_2O$. (a) The proton signal intensity from rapid exchange measurement data normalized to remove $T_1$ effects at different mixing times. (b) Exchanging fractions from rapid measurements with $t_m = 0.2$ (blue), 4 (red), and 20 ms (orange). (c) 1-D diffusion measurements were performed at points throughout the timecourse (seen as breaks in the data in (a)) and distributions are presented as signal fractions ($P(D) \times I_0$).

## Appendix 4

### Penetration timescales

The methods were used to follow penetration of deuterated water $D_2O$ into the tissue as well as the delipidation of the tissue via Triton X. For the $D_2O$ wash (**Appendix 3—figure 1**) $I_0$ decreased to 0.36 two minutes after washing, indicating that water in the tissue communicates with the aCSF on timescales of minutes. Exchange and restricted fractions decreased during delipidation on the timescale of roughly one day (**Figure 7**). Mass transport theory estimates the timescale to equilibrate a concentration gradient across the tissue as $\approx$ 1 min for water and $\approx$ 3 hrs for Triton X based on $t = r^2/4D$ (**Crank, 1979**) with specimen radius $r = 0.7$ mm and measured $D = 2.15 \times 10^{-9}$ m$^2$/s for water and $D = 1.3 \times 10^{-11}$ m$^2$/s for Triton X (see **Figure 7—figure supplement 2**). The timescale of water penetration was consistent with mass transport theory (**Crank, 1979**), but the timescale of Triton X penetration and delipidation was longer than predicted. An increased time is expected due to the reaction front which develops as Triton X delipidates, slowing its overall penetration.

# Appendix 5

## Supplementary materials and methods

### Test chamber

The bottom portion of the experimental test chamber was made of aluminum to provide good heat conduction to the media. A bored-out rectangular hole with a glass cover slide glued to the bottom held the media, solenoid, and spinal cord. Aluminum parts that contacted aCSF were coated with a thin layer of RTV silicone to avoid corrosion. The top of the chamber was made from poly(methyl methacrylate) (PMMA) with two inlets for inflow and outflow of the gas and a hole for the PicoM fiber optic temperature sensor (Opsens Solutions Inc, Québec, Canada). Temperature measurements were not affected by RF and did not induce noise in the RF system. The temperature of the bath was monitored and recorded continuously. The top slide was secured and mounted to the bottom part with four screws. The NMR solenoid coil was glued to the glass cover slip bottom with a hot glue gun. Two separate chambers and solenoid coils were built for live and fixed spinal cord specimen.

### NMR hardware

The solenoid radiofrequency (RF) coils and the circuit were built in-house. Solenoids were made from wrapping two concentric layers of AWG 30 copper wire around a #2–56 plastic screw totaling 39 turns resulting in 2 mm inner diameter 4 mm outer diameter, 1.3 cm length. The resulting coils had an inductance $L \sim 600$ nH and impedance $X \sim 52$ $\Omega$ at 13.79 MHz.

The solenoid connected to a circuit board with detachable pin connectors. The circuit design is shown in *Appendix 5—figure 1.* The circuit used two trimmer capacitors (NMAJ25HV, Knowles Voltronics) with tunable range 1–23 pF for tune and match. The circuit board was connected to the Kea2 spectrometer by a 50 $\Omega$ coax cable. The coil matched to $-34$ dB at 13.79 MHz when immersed in aCSF. RF pulses were driven by a 100 W RF pulse amplifier (Tomco, Adelaide, Australia).

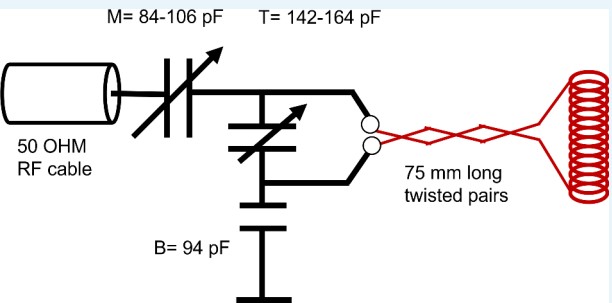

**Appendix 5—figure 1.** RF circuit design. Drawing of the circuit for the RF showing the capacitance of the tune (T), match (M), and balance (B). Note a single wrapping of the solenoid was drawn rather than the actual double-wrap for visual simplicity.

### 1-D spin echo diffusion

A standard pulse sequence (SEdec in Prospa) was used for measuring diffusion with a static gradient (*Rata et al., 2006*) built off of a spin echo followed by a Carr–Purcell–Meiboom–Gill (CPMG) echo train (*Hahn, 1950*; *Carr and Purcell, 1954*; *Meiboom and Gill, 1958*), as shown in *Appendix 5—figure 2a*. The phase cycle list was four scans long (*Casanova et al., 2011*).

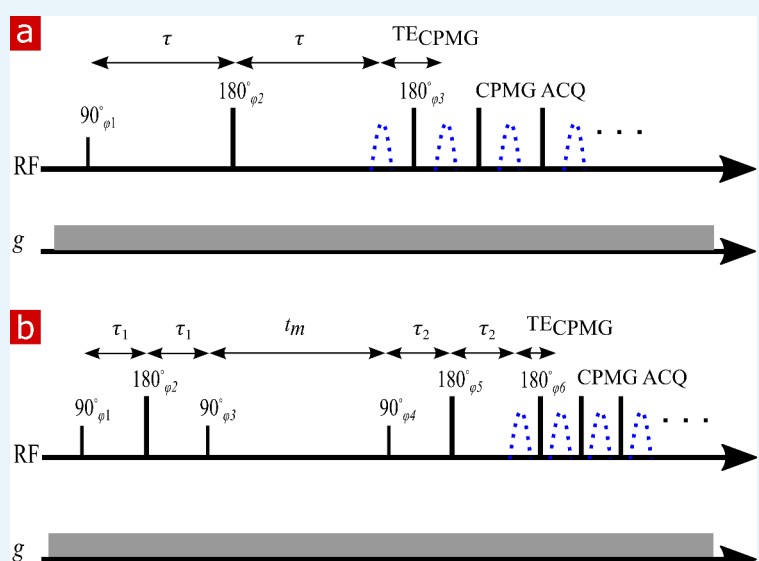

**Appendix 5—figure 2.** NMR diffusion and exchange pulse sequences. (**a**) Spin echo pulse sequence for measuring diffusion with a static gradient. $\tau$ is varied to control $b$. (**b**) DEXSY pulse sequence for measuring exchange with a static gradient. The two SE encoding blocks with $\tau_1$ and $\tau_2$, varied independently to control $b_1$ and $b_2$, are separated by $t_m$. Signal is acquired in a CPMG train for both (**a**) and (**b**).

Signal from the CPMG train is summed up as one data point. This summation provides a significant boost in the signal-to-noise ratio (SNR). Only signal from the real channel (rather than signal magnitude) is taken. This leads to zero-mean Gaussian rather than Rician noise, a significant benefit for multiexponential signal analysis. The echo time of the spin echo $\tau$ is incremented in successive loops of the experiment to encode for diffusion. For water and with a strong static gradient, the attenuation due to $T_2$ relaxation during $\tau$ is insignificant. Signal can be modeled using **Equation 1** for pure liquids such as water. In heterogeneous materials such as biological tissue, water in different parts of the material experience different hindrances and restrictions. Each sub-ensemble of water molecules has its own effective self-diffusion coefficients. The signal can be modeled as arising from the distribution of effective or apparent self-diffusion coefficients of the water in the different environments using

$$I(b)/I_0 = \int_0^\infty P(D)e^{-bD}dD. \tag{6}$$

## DEXSY pulse sequence

**Appendix 5—figure 2b** shows the static gradient spin echo DEXSY pulse sequence. In this sequence, molecules are encoded for their diffusion coefficient in their local environment during the first interval $\tau_1$. Magnetization is then stored for a mixing time, $t_m$ during which time molecules move freely and may exchange between diffusive environments. (Note that this definition of $t_m$ is like the definition used by **Washburn and Callaghan (2006)** for relaxation exchange spectroscopy (REXSY or $T_2 - T_2$), and is different from the original definition presented by **Callaghan and Furó (2004)** for pulsed gradient spin echo DEXSY which included the gradient pulse duration. We choose not to use the **Callaghan and Furó (2004)** definition since $\tau$ is changing in a static gradient spin echo DEXSY measurement whereas the definition of $t_m$ should require that it be constant throughout the whole experiment.) Molecules are again encoded for diffusion during $\tau_2$ and then signal is acquired in a CPMG train.

## Full DEXSY

The full 2-D DEXSY can be analyzed as a diffusion exchange distribution, related to the signal through

$$I(b_1, b_2) = \int_0^\infty \int_0^\infty P(D_1, D_2) e^{-b_1 D_1 - b_2 D_2} dD_1 dD_2, \tag{7}$$

a 2-D version of **Equation 6**. The diffusion encoding variables, $b_1$ and $b_2$ are varied by independently varying $\tau_1$ and $\tau_2$. Molecules which do not exchange between environments will have the same diffusion coefficient during $\tau_1$ and $\tau_2$ contributing to populations on the diagonal of the 2-D distribution (see, e.g., Figure 2a and b in **Cai et al., 2018**). Molecules which do exchange and thus are encoded with different diffusion coefficients between the two $\tau$ will contribute to off-diagonal exchange peaks in the 2-D distribution.

## Rapid exchange measurement

Alternatively to the full DEXSY, we recently introduced a rapid method for measuring exchanging fractions. The measurement relies on curvature of the raw DEXSY signal after a variable transformation, first shown by **Song et al. (2016)** for REXSY. In particular, this method shows that exchange between diffusion environments results in the raw data being curved up along a slice of constant $b_s = b_1 + b_2$ (see Figure 2c and d in **Cai et al., 2018**). The exchanging fraction scales with $(\partial^2 I / \partial b_d^2)$ where $b_d = b_2 - b_1$. The second derivative can be approximated with the 2nd order finite difference method,

$$\frac{\partial^2 I}{\partial b_d^2}\Big|_{b_d=b} \approx \frac{I|_{b_d=b-\Delta b_d} - 2I|_{b_d=b} + I|_{b_d=b+\Delta b_d}}{\Delta b_d^2}, \tag{8}$$

omitting higher order terms. The greatest sensitivity to exchange is when the central point is acquired at $b_d = 0$ and the edges are acquired at $b_d = \pm b_s$. Normalizing by a datapoint acquired with no diffusion weighting $b_s = 0$ removes relaxation effects. This is a relative measure of exchange and is enough to provide image contrast in MRI, to look at time-varying processes, or to measure exchange rates (discussed below).

In **Cai et al. (2018)** we developed the theory for obtaining the exchanging fraction, $f$, from $(\partial^2 I / \partial b_d^2)$, which for a two-site exchange model results in

$$\begin{aligned} f &= \left( \frac{\partial^2 I}{\partial b_d^2}\Big|_{b_d=b} \right) \frac{e^{b_s D_s}}{\cosh(b_d D_d) D_d^2} \\ &= \left( \frac{\partial^2 I}{\partial b_d^2}\Big|_{b_d=0} \right) \frac{e^{b_s D_s}}{D_d^2} \end{aligned} \tag{9}$$

where

$$D_s = \frac{(D_e + D_i)}{2}; \quad D_d = \frac{(D_e - D_i)}{2}. \tag{10}$$

For a true two site system for which each component attenuates by $\exp(bD)$, $D_e$ and $D_i$ can be measured from a biexponential fit. However, the spinal cord system under study was multiexponential over the entire $b$ range such that the measurement of $D_e$ and $D_i$ was ambiguous and prone to the same challenges discussed regarding fitting and interpreting 1-D diffusion data for neural tissue. Rather than attempting to measure $D_e$ and $D_i$ for each sample and being prone to misinterperetation, the values were fixed at $D_e = 10^{-9}$ and $D_i = 10^{-11} \ m^2/s$. In this way we explicitly acknowledge that measured values of $f$ are only relative and not absolute. Fortunately, measured AXRs are independent of the choice of $D_e$ and $D_i$.

## DEXSY sequence phase cycles

Attention was paid to phase cycles for the static gradient spin echo DEXSY sequence (**Appendix 5—figure 2**) due to each RF pulse being imperfect and exciting multiple coherence pathways when the inhomogeneity of the magnetic field is greater than the bandwidth of the RF pulses (**Hürlimann, 2001**). The phase cycle list was eight scans long and is shown in **Appendix 5—table 1**.

**Appendix 5—table 1.** Static gradient spin echo DEXSY Phase Cycles.

| $\varphi_1$ | $\varphi_2$ | $\varphi_3$ | $\varphi_4$ | $\varphi_5$ | $\varphi_6$ | $\varphi_{rec}$ |
|---|---|---|---|---|---|---|
| 0 | $+\pi/2$ | 0 | 0 | $+\pi/2$ | $\pi/2$ | $\pi$ |
| $\pi$ | $-\pi/2$ | 0 | 0 | $+\pi/2$ | $\pi/2$ | 0 |
| 0 | $+\pi/2$ | $\pi$ | 0 | $+\pi/2$ | $\pi/2$ | 0 |
| $\pi$ | $-\pi/2$ | $\pi$ | 0 | $+\pi/2$ | $\pi/2$ | $\pi$ |
| 0 | $+\pi/2$ | 0 | $\pi$ | $-\pi/2$ | $\pi/2$ | 0 |
| $\pi$ | $-\pi/2$ | 0 | $\pi$ | $-\pi/2$ | $\pi/2$ | $\pi$ |
| 0 | $+\pi/2$ | $\pi$ | $\pi$ | $-\pi/2$ | $\pi/2$ | $\pi$ |
| $\pi$ | $-\pi/2$ | $\pi$ | $\pi$ | $-\pi/2$ | $\pi/2$ | 0 |

Although the phase cycle list is not exhaustive, we found the signal to be well-behaved on a non-exchanging two-pool system comprised of a capillary filled with polydymethylsyloxane bathed in water. Signal as a function of $b_1$ or $b_2$ were symmetric and decayed the same as signal as a function of $b$ from the 1-D SEdec sequence. Additionally, signal was flat along slices of constant $b_s$ and the 2-D DEXSY map showed two diffusion coefficients along the $D_1 = D_2$ diagonal equal to $D_{\mathrm{water}}$ and $D_{\mathrm{PDMS}}$. One exception was that the phase cycles let through signals which do not form a gradient echo when they see the storage pulse, but do form a gradient echo upon acquisition, thus seeing the sequence as a 1-D stimulated echo diffusion. This became apparent due to additional refocussing when $b_1 = b_2$. This was found to be an issue with a previous miniature flat RF coil design (**Bai et al., 2015**) but went away when switching to the solenoid RF coil, thus it is likely an issue of $\mathrm{B}_1$ inhomogeneity of the miniature flat RF coil (**Watzlaw et al., 2013**). To avoid refocussing this signal, points were never acquired exactly on $b_1 = b_2$. Also note that the static gradient acts as a crusher during the storage interval. The sequence selects both compensated and uncompensated signals (**Khrapitchev and Callaghan, 2001**). The sequence can be compared and contrasted to another DEXSY sequence developed by **Neudert et al. (2011)** for static gradients but using stimulated echoes for diffusion encoding.

## Fitting exchange parameters

Exchange rates can be estimated from the full DEXSY or the rapid measurement by repeating the measurement with multiple mixing times which span the exchange process. Exchange parameters can be determined assuming exchange between diffusion environments is governed by a first order rate law of the form $df_{i,e}/dt = k_{i,e}f_{i,i} - k_{e,i}f_{e,i}$ with rate constants $k_{i,e}$ and $k_{e,i}$ (**Washburn and Callaghan, 2006**). The data we present calls for a nonzero initial condition; $f_{i,e}(t_m = 0) = f_{i,e0}$ (discussed below). The resulting two-site exchange model is:

$$f_{i,e}(t) = f_{e,i}(t) = \frac{f(t)}{2} = \left( \frac{f_e k_{e,i}}{k_{e,i} + k_{i,e}} - f_{e,i0} \right)\left(1 - e^{-(k_{i,e}+k_{e,i})t}\right) + f_{e,i0}$$

$$= \left( \frac{f_i k_{i,e}}{k_{i,e} + k_{e,i}} - f_{i,e0} \right)\left(1 - e^{-(k_{e,i}+k_{i,e})t}\right) + f_{i,e0},$$

(11)

with equilibrium fractions $f_e$ and $f_i$. With either the full DEXSY or the rapid measurement, exchanging fractions as a function of mixing time can be fit with a 3-parameter model of the form

$$f(t_m) = (f_{SS} - f_0)[1 - e^{-kt_m}] + f_0$$

(12)

to estimate the initial exchange fraction $f_0 = 2f_{i,e0}$, the steady-state exchange fraction $f_{SS} = \frac{2f_i k_{i,e}}{k_{i,e} + k_{e,i}}$, and the characteristic exchange rate $k = k_{i,e} + k_{e,i}$ (called the apparent exchange rate, AXR, in the text).

The data we present calls for a nonzero initial condition. A previous study of $T_2$–$T_2$ exchange in a polymer–solvent system near the glass transition also observed finite exchange

when $t_m \approx 0$ (*Williamson et al., 2019*). Models have shown significant exchange during the encoding periods $\tau$ can lead to exchange peaks at $t_m = 0$ (*Schwartz et al., 2013*).

## Appendix 6

### Representative relaxation time measurements

Spin-lattice ($T_1$) relaxation is the timescale for magnetization of the sample to equilibrate with the external magnetic field and occurs in the vector longitudinal to the applied $\mathrm{B}_0$ field. Spin-spin ($T_2$) relaxation is the timescale for the the spin magnetization to dephase and occurs in the plane transverse to $\mathrm{B}_0$. Magnetization is always undergoing either $T_1$ or $T_2$ relaxation during an NMR experiment and the relaxation times are therefore pertinent information when designing an NMR protocol. (Relaxation time values averaged across groups are presented in Materials and methods subsection NMR experimental methods. *Appendix 6—figure 1* shows a representative $T_2$ measurement performed on a fixed spinal cord. Note that $T_2$ is measured under the static gradient such that decay is also occurring due to diffusion, although diffusive decay is minimized bu using a minimum echo time ($\mathrm{TE} = 25 \ \mu s$) (*Carr and Purcell, 1954*). An important observation is the lack of a rapid (1 ms timescale) initial decay. Hence, $T_2$ negligibly impacts the attenuation in spin echo diffusion measurements. *Appendix 6—figure 2* shows a representative $T_1$ measurement performed on a fixed spinal cord. The $T_1$ is significantly longer than the longest mixing time used in exchange measurements.

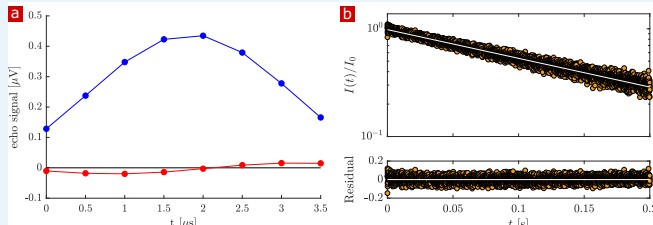

**Appendix 6—figure 1.** $T_2$ measurement. Representative Carr-Purcell-Meiboom-Gill (CPMG) (10 s repetition time (TR), 8000 echoes, TE = 25 μs) on fixed spinal cord. (**a**) The echo shape summed over all echoes (real signal (blue) phased maximum and imaginary signal phased to zero (red)). (**b**) Real signal decay (orange circles) and exponential fit with $T_2 = 163$ ms and $I_0 = 6.5 \mu \mathrm{V}$ (white line) and residuals of the fit.

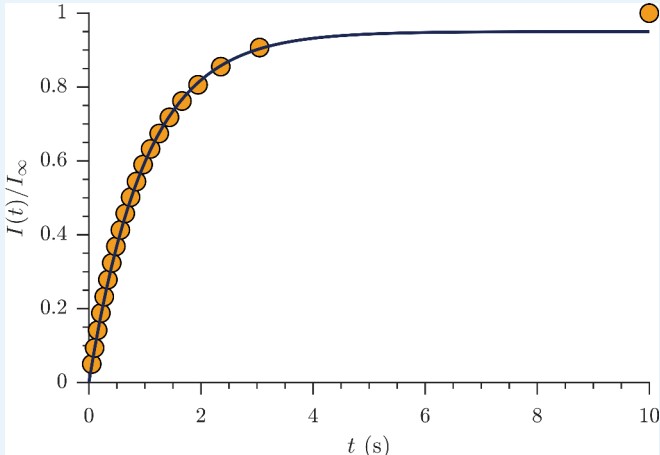

**Appendix 6—figure 2.** $T_1$ measurement. Representative saturation recovery experiment on fixed spinal cord (1 s TR, 21 recovery time points logarithmically spaced from 50 ms to 10 s) showing signal intensity normalized by signal at 10 s recovery time (orange circles) and exponential fit with estimated $T_1$ = 990 ms (solid black line).

## Appendix 7

### Rapid exchange measurement tests

Given that this is the first time the rapid exchange measurement has been used with a static gradient as well as on anything other than an ideal phantom, a full characterization of the signal seemed necessary. Details of the rapid exchange measurement method can be found in **Cai et al. (2018)**, and Appendix 5. In this section we test the behavior of the signal acquired as a function of $b_d$, $b_s$, and $t_m$ on fixed spinal cord and compare results to the predicted exchange behavior (**Cai et al., 2018**).

The curvature along slices of $b_s = 4500 \ \mathrm{s/mm^2}$ as a function of $b_d$ at different mixing times is shown in **Appendix 7—figure 1**. The signal is concave up with maximum at $b_d = \pm 4500$, minimum at $b_d = 0$, and roughly symmetric about $b_d = 0$, as expected. Exchange increases with mixing time, also as expected. From this, we can conclude that the 4-point method (discussed in Appendix 1) can capture the exchange with maximal sensitivity.

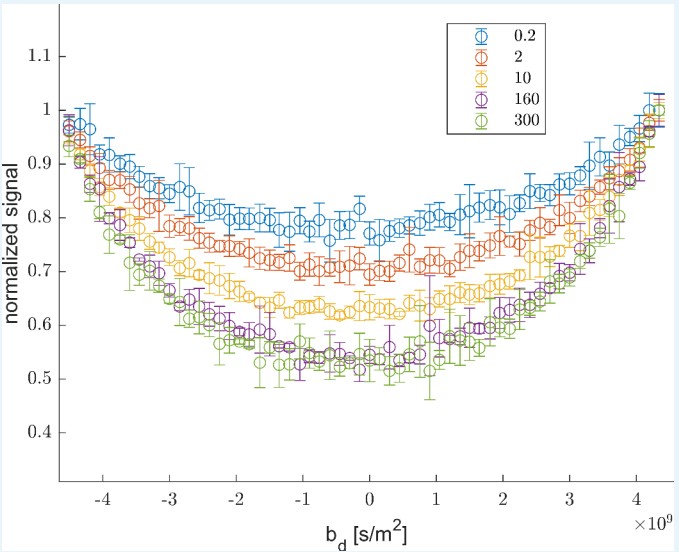

**Appendix 7—figure 1.** Rapid exchange test #1. The curvature along slices of $b_s = 4500 \ \mathrm{s/mm^2}$ as a function of $b_d$ highlighting the increase in the depth of the curvature with increasing $t_m$, shown in figure legend. The increased depth is due to increased exchange (**Song et al., 2016**; **Cai et al., 2018**).

The optimal $b_s$ maximizes the finite difference (Appendix 5 **Equation 8**) and provides optimal sensitivity to exchange in the presence of noise. However, different $b_s$ values may diffusion-weight the measurement towards exchange between different pools. **Appendix 7—figure 2** shows that the finite difference reaches a maximum near $b_s = 6000 \ \mathrm{s/mm^2}$. The value $b_d = 4500 \ \mathrm{s/mm^2}$ used in this study is thus near the optimum. Additionally, from this data it was found that the exchange rate is not significantly different between $b_s$ values, indicating that AXR is not very sensitive to $b_s$ for spinal cord samples.

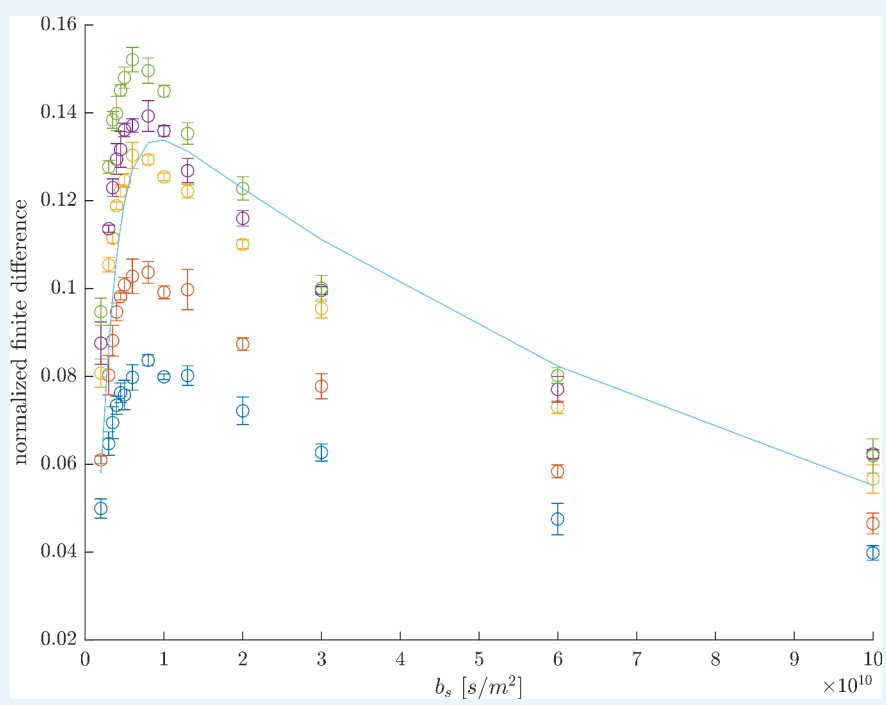

**Appendix 7—figure 2.** Rapid exchange test #2. Difference between $2I_{b_d=0}$ and $I_{bd=+bs} + I_{bd=-bs}$, normalized by $I_{0,0}$, as a function of $b_s$, showing the optimal $b_s$ for measuring the largest curvature response as occurring near $b_s = 6000\ \mathrm{s/mm^2}$. The line is a prediction of the finite difference for a two-site system, from Equation 8 in *Cai et al. (2018)* using $f = 0.15$, $D_e = 10^{-9}$, and $D_i = 10^{-11}\ \mathrm{s/mm^2}$.

The four-point method allowed for high temporal resolution of exchange. *Appendix 7—figure 3* shows a dense sampling of $f$ as a function of $t_m$, acquired overnight on a fixed spinal cord specimen. The data shows the $t_m = 0.1$ ms point to be not well behaved, with $f$ decreasing from $t_m = 0.1$ to $0.2$ ms, but increases from $t_m = 0.2$ ms onwards, indicating that $t_m = 0.2$ ms is a good point for the minimum $t_m$. The exchange plateaus near $t_m = 100$ ms and stays roughly constant to $t_m = 300$ ms, indicating that $t_m = 300$ ms is a good point for the maximum $t_m$ because it captures the maximum, steady state exchange, and it does not show $T_1$ relaxation effects (in particular due to differences in $T_1$ between exchanging pools as discussed in *Cai et al. (2018)*. The data was fit with the first order rate model, *Equation 12*, to estimate AXR. The dense sampling shows that the data is not fully explained by the first order rate model; it rises up quicker and plateaus slower. However, rather than fitting a model with more parameters, for example a model with two AXRs, we choose to stick to the first order rate model with one AXR. The estimated AXR is not statistically different from the values measured from the AXR measured from the standard 11 $t_m$ point protocol used throughout the text, indicating that the 11 point protocol does not bias the measured AXR. The dense sampling also iterates $f(t_m = 0) \neq 0$, thus calling for a non-zero initial condition (discussed in Appendix 5).

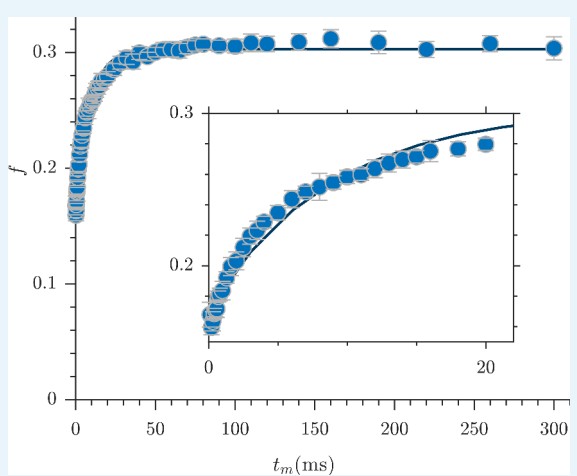

**Appendix 7—figure 3.** Rapid exchange test #3. High temporal resolution exchange rate measurement from the rapid measurement with $b_s = 4500 \ \mathrm{s/mm^2}$ and 53 $t_m$ points between 0.1 and 300 ms. First order rate model fits estimate AXR = $113.7 \pm 5.2 \ \mathrm{s^{-1}}$ (solid line).

