## [Decision Letter]

**Acceptance summary:**

This work demonstrates the ability of diffusion magnetic resonance to detect and measure permeability of membrane structures smaller than the cell. The development of the method to non-invasively measure water diffusion and exchange in sub-cellular structures makes may enable imaging of normal changes in organelle structure and permeability within living tissue for basic biology and medicine.

**Decision letter after peer review:**

Thank you for submitting your article "Magnetic resonance measurements of cellular and sub-cellular membrane structures in live and fixed neural tissue" for consideration by *eLife*. Your article has been reviewed by two peer reviewers, and the evaluation has been overseen by a Reviewing Editor and Christian Büchel as the Senior Editor. The following individuals involved in review of your submission have agreed to reveal their identity: Valerij Kiselev (Reviewer #2).

The reviewers have discussed the reviews with one another and the Reviewing Editor has drafted this decision to help you prepare a revised submission.

Summary:

The authors present a novel technology and experimental setup to measure water molecules restricted diffusion in the range 200-1400 nm attributed to small cellular organelles. The authors designed record-breaking experiments in which they have reached a previously unavailable range of measurement parameters. This paper is a nice contribution with potential new understanding on the biological sources of diffusion MRI signal. It confirms the previous work by Leuze et al. (Neuroimage 2017) suggesting the source for diffusion anisotropy (and restricted diffusion) is the membrane and lipids in the tissue. However, the statements such as "Low-field single-sided Magnetic Resonance diffusion methods detect and measure permeability of sub-micron compartments which are likely organelles and cellular vesicles within ex-vivo mouse spinal cords" needs more thorough validation. Moreover, it would be beneficial for the authors to present their results on single diffusion weighting as mainly experimental, open to future theoretical interpretations. Such an approach is common in other scientific disciplines and we believe that NMR is mature enough to adapt this approach, at least when addressing such a complicated problem as diffusion in living tissues.

Essential revisions:

1) Reproducibility: This work is based on an unknown number of samples. In the results the number six (later on becomes five) for fixated samples appears but it is not mentioned anywhere in the Materials and methods. Nine samples are reported somewhere for live tissues but again – not explicitly mentioned in the Material and methods. In the Materials and methods, however, the number n=2 is mentioned for post Triton treatment. I think the samples need to be explicitly mentioned and described in the Materials and methods. If indeed the numbers were 9/6/5/2 – a proper statistical analysis for reproducibility is required to demonstrate the strength of the results.

2) Why did the authors choose postnatal day 4 spinal cord, and not a mature spinal cord? It does not seem to be a trivial choice.

3) Diffusion time is an important factor in conventional DTI with many implications on the measured signal. While the diffusion encoding time (τ) reflects somehow the diffusion time, its interpretation is different from the diffusion time. I think that for a general journal such as *eLife* – a better description of the jargon is needed to avoid confusion.

4) On the same subject, restricted diffusion is estimated by a diffusion time dependent experiment (as was shown several times by the senior author). How one can estimate restricted diffusion from a diffusion exchange experiment? The notion "restricted diffusion" appears throughout the paper but without an experiment that proved that (at least to my understanding). DEXSY measures exchange of specific water pool – not restricted diffusion. The fact that a certain population of water molecules has a slow diffusion coefficient within an exchange distance of 200-1400nm- does not mean it is restricted (maybe it is just bound with slow motion?). I think this issue needs to be clarified.

5) The authors note that ACSF diffusivity should be a delta function but eventually turned to be a broad distribution because of a regularization procedure that is required by the inversion analysis. While that might be the case, could it also be that this is due to in-homogeneous gradients? Is it possible to control somehow to the effect of the regularization (maybe perform a more conventional analysis?) What is the effect of such regularization on the slow-diffusing components (those who claimed to be arising from organelles)?

6) The 2D DEXSI experiment is the main result of this paper. In Figure 3 the authors note: "The distribution is divided into a 3x3 grid for the possible exchange pathways between components A (3.2×10−4 −1×10−2), B(1×10−2 −3.2×10−1), and C (5.6 × 10−1 − 1 × 101 D/D0), shown by the color coding and labels." – what is the basis of this division? Just to guide the eye? Was some cluster analysis involved in such division?

7) As this is a constant gradient experiment, there is no indication of rotational invariance of the results. I think this is extremely important – especially since the authors claim to reveal restricted diffusion within organelles, which must be separated from the main source of restricted diffusion in neural tissue – the neural fiber. This issue deserves, at least, to be mentioned in the Discussion.

8) Sampling the domain of ultrahigh diffusion weighting using the single diffusion encoding (Figure 2). This is a very interesting part, which is re-iterated in Discussion with the conclusion about the sub-micrometer-sized water confinement. This conclusion is based on a quantitative analysis, which seems to be applied beyond its validity range, which is the main concern about this manuscript.

9) Another concern is about the interpretation of the numerical inverse Laplace transformation, which is an adequate signal description in a very limited range of parameters. Although the authors declare the usage as a representation, "which makes no assumptions about tissue microstructure", they actually rely on the quantitative outcome of this transformation, in particular, for one of the main finding about the 25% of water restricted by membranes.

10) Studying exchange using two diffusion encoding separated by a mixing time. This is the central part of the study, which - in a clever way - decouples from the problem associated with the previous part.

11) Studying the origin of restrictions using the substitution with deuterated water and the tissue delipidation – this part should have a high impact on the further development in the field.

---

## [Author Response]

Summary:The authors present a novel technology and experimental setup to measure water molecules restricted diffusion in the range 200-1400 nm attributed to small cellular organelles. The authors designed record-breaking experiments in which they have reached a previously unavailable range of measurement parameters. This paper is a nice contribution with potential new understanding on the biological sources of diffusion MRI signal. It confirms the previous work by Leuze et al. (Neuroimage 2017) suggesting the source for diffusion anisotropy (and restricted diffusion) is the membrane and lipids in the tissue. However, the statements such as "Low-field single-sided Magnetic Resonance diffusion methods detect and measure permeability of sub-micron compartments which are likely organelles and cellular vesicles within ex-vivo mouse spinal cords" needs more thorough validation. Moreover, it would be beneficial for the authors to present their results on single diffusion weighting as mainly experimental, open to future theoretical interpretations. Such an approach is common in other scientific disciplines and we believe that NMR is mature enough to adapt this approach, at least when addressing such a complicated problem as diffusion in living tissues.

We have changed “are likely” to “likely include”. We have added text to point out that interpretation and modeling efforts are ongoing, and that our methods and the data we have obtained can assist in these efforts.

“Interpretation and modeling of the signal attenuation from diffusion measurements on neural tissue is an ongoing area of research (Novikov et al., 2018). Nonparametric data inversion techniques can model signal attenuation as arising from distributions of diffusion coefficients (Pfeuffer et al., 1999).”

“The estimate of restriction length *R* uses a theoretical model for the motional averaging regime (Equation 5) and assumes that *R<l_g_* such that spins can diffuse across the restriction many times without significant dephasing. […] Nonetheless, the interplay between the attenuation regimes, heterogeneity of restriction sizes, and exchange make the modeling of diffusive motion of water in biological tissue a very challenging problem, and an important topic for future research which before now could not be experimentally studied or validated.”

Essential revisions:1) Reproducibility: This work is based on an unknown number of samples. In the results the number six (later on becomes five) for fixated samples appears but it is not mentioned anywhere in the Materials and methods. Nine samples are reported somewhere for live tissues but again – not explicitly mentioned in the Materials and methods. In the Materials and methods, however, the number n=2 is mentioned for post Triton treatment. I think the samples need to be explicitly mentioned and described in the Materials and methods. If indeed the numbers were 9/6/5/2 – a proper statistical analysis for reproducibility is required to demonstrate the strength of the results.

We have added a subsection to the Materials and methods section titled “Statistical analysis and reproducibility” to make clear the ways in which we have quantified repeatability. The text is also reproduced here.

“The Results section presents data from multiple measurements repeated on individual specimen as well as measurements performed on groups of samples with different treatments (n = x). […] Means and standard deviations are presented to quantify repeat measurement and sample-to-sample reproducibility of the results.”

We added text regarding repeat measurement reproducibility.

“The signal intensity from 30 measurements performed on the same sample over the course of 30 hours varied similarly to the background noise (Figure 2—figure supplement 1). Measurement variability is thus simply determined by SNR and the system is amenable to long scans and signal averaging.”

We have added Figure 2—figure supplement 1, Figure 4—figure supplement 1, and Figure 6—figure supplement 1 which provide descriptive statistics regarding repeat measurement and sample-to-sample reproducibility of the results. These statistics can be utilized in power analyses to guide future experimental designs.

2) Why did the authors choose postnatal day 4 spinal cord, and not a mature spinal cord? It does not seem to be a trivial choice.

We have added the following text:

“Neonatal mice were studied rather than adult mice because viability of ex vivo spinal cords is known to decrease with age (Fulton, 1986). Ex vivo Spinal cords become more prone to hypoxia as they grow larger and oxygen in the tissue is consumed faster than it can diffuse to the center (Wilson, Chersa and Whelan, 2003). We directly demonstrate that spinal cords are alive after hours of NMR measurements by recording electrical responses from motoneurons.”

3) Diffusion time is an important factor in conventional DTI with many implications on the measured signal. While the diffusion encoding time (τ) reflects somehow the diffusion time, its interpretation is different from the diffusion time. I think that for a general journal such as eLife – a better description of the jargon is needed to avoid confusion.

We have added the following text

“The displacements contain averaged information about the hindrances and restrictions which the molecules experienced during their random trajectories through the microstructure. […]Both methods lead to a measured signal attenuation, an effect which can be summarized in a single variable, b (Stejskal and Tanner, 1965).”

4) On the same subject, restricted diffusion is estimated by a diffusion time dependent experiment (as was shown several times by the senior author). How one can estimate restricted diffusion from a diffusion exchange experiment? The notion "restricted diffusion" appears throughout the paper but without an experiment that proved that (at least to my understanding). DEXSY measures exchange of specific water pool – not restricted diffusion. The fact that a certain population of water molecules has a slow diffusion coefficient within an exchange distance of 200-1400nm- does not mean it is restricted (maybe it is just bound with slow motion?). I think this issue needs to be clarified.

We are very confident with our interpretation of restriction, however we understand that we need clarify and discuss this. We utilize signal decay based definitions of restriction from NMR literature (e.g., Hurlimann et al., 1995). We define restriction as being on the timescale of the measurement. Indeed, research with pulsed gradient methods show that in biological tissue all water communicates on long enough time scales. However, we define restriction as being on the timescale of the measurement. Additionally, the models presented for restriction point out how drastically the signal attenuation deviates from that of free or hindered diffusion (shown in Figure 5). This drastic difference leads us to believe that our measurement effectively separates water, which is restricted from water which is freely diffusing on the timescale of the measurement. We have defined restricted diffusion and restricted water more clearly and we have added text to the Discussion.

“The single-sided magnet's field strength decreases rapidly with distance from the top surface, with a gradient of g=15.3 T/m. […] Alternatively from diffusion coefficient distribution analysis, integrals of P(D/D0) on either side of D/D0=0.17 are heuristically used as measures of the free and restricted water fraction.”

“Diffusion measurements with a large static gradient separate free water from restricted water because freely diffusing water attenuates exponentially with b whereas water which feels the boundaries of membranes attenuates exponentially with b^1/3^ (Grebenkov, 2018) (which is proportional to τ, as used for the abscissa in Figure 2) after the free water component has attenuated. The linearity of the attenuation at long τ (in Figure 2) provides an additional signature of restricted water.”

5) The authors note that ACSF diffusivity should be a delta function but eventually turned to be a broad distribution because of a regularization procedure that is required by the inversion analysis. While that might be the case, could it also be that this is due to in-homogeneous gradients? Is it possible to control somehow to the effect of the regularization (maybe perform a more conventional analysis?) What is the effect of such regularization on the slow-diffusing components (those who claimed to be arising from organelles)?

Whether or not the gradient is constant over the active region is a valid question, particularly for such inhomogeneous low-field systems. However, it appears that the NMR-MOUSE is so inhomogeneous that the gradient is very well-defined and constant at the chosen frequency. Our evidence is from how well a single exponential attenuation model fits the diffusion data of aCSF. This is very clear in Figure 1—figure supplement 1. We have added the following sentence to the main text.

“Error residuals are random with variance consistent with the noise of the system (Figure 1—figure supplement 1).”

Choosing the amount of regularization is an important point which we put a lot of thought into. The regularization procedure is already stated in the NMR data analysis subsection of the Materials and methods. With regards to a more conventional analysis, utilizing the raw signal is in a way the most conventional analysis, as this is diffusion weighting in its rawest form. We have utilized both raw signal and the inverted distributions in making observations. We do not claim that any particular peak on the distribution of diffusion coefficients is associated with organelles. However, we do feel that components get spread out along this distribution based on their apparent mobility. We have added the following text to make readers aware of the nature of the inversion of diffusion signal attenuation data.

“Data inversion methods used to obtain 1- and 2-D distributions of diffusion coefficients assume that the data attenuates exponentially with b. […] New development of nonparametric diffusion models may allow further interpretations.”

6) The 2D DEXSI experiment is the main result of this paper. In Figure 3 the authors note: "The distribution is divided into a 3x3 grid for the possible exchange pathways between components A (3.2×10−4 −1×10−2), B(1×10−2 −3.2×10−1), and C (5.6 × 10−1 − 1 × 101 D/D0), shown by the color coding and labels." – what is the basis of this division? Just to guide the eye? Was some cluster analysis involved in such division?

“This division was chosen in an attempt to separate the free water component (C) from the restricted water component, and to separate the restricted component into two groups (B and A) based on their apparent mobility.”

7) As this is a constant gradient experiment, there is no indication of rotational invariance of the results. I think this is extremely important – especially since the authors claim to reveal restricted diffusion within organelles, which must be separated from the main source of restricted diffusion in neural tissue – the neural fiber. This issue deserves, at least, to be mentioned in the Discussion.

This is a valid point which in the interest of brevity we will leave for a future study. To make the reader more aware of the directionality of the diffusion measurement relative to the fiber, we have added

“Solenoid radiofrequency (RF) coils were specially built to the size of the spinal cords under study. In the solenoid coil, the spinal cord is oriented with its length perpendicular to the gradient such that the system measures diffusion of water perpendicular to the spinal cord.”

To highlight the need for future study to separate the contributions to the restricted water signal from the cellular processes and from organelles, we have added

“The fixed direction of the gradient relative to the sample inhibits the study of anisotropy. […]A combined study would be additionally beneficial by broadening the window of resolvable structure sizes (Benjamini et al., 2014, Benjamini et al., 2016, Nilsson, 2017).”

8) Sampling the domain of ultrahigh diffusion weighting using the single diffusion encoding (Figure 2). This is a very interesting part, which is re-iterated in Discussion with the conclusion about the sub-micrometer-sized water confinement. This conclusion is based on a quantitative analysis, which seems to be applied beyond its validity range, which is the main concern about this manuscript.

We agree that this is a very interesting part of the paper. We also agree that the novelty of this data for such a range of diffusion weightings somewhat implies that current models need to be tested and perhaps new models need to be developed and tested. We hope that our experimental data will facilitate research in this area. We have added the following text to let people less familiar with the diffusion modeling understand that the estimates of restriction size are approximate and to appeal to the modeling audience and push them towards what we believe to be an exciting area of research.

“The estimate of restriction length R uses a theoretical model for the motional averaging regime (Equation 5) and assumes that R< l_g_ such that spins can diffuse across the restriction many times without significant dephasing. […] Nonetheless, the interplay between the attenuation regimes, heterogeneity of restriction sizes, and exchange make the modeling of diffusive motion of water in biological tissue a very challenging problem, and an important topic for future research which before now could not be experimentally studied or validated.”

9) Another concern is about the interpretation of the numerical inverse Laplace transformation, which is an adequate signal description in a very limited range of parameters. Although the authors declare the usage as a representation, "which makes no assumptions about tissue microstructure", they actually rely on the quantitative outcome of this transformation, in particular, for one of the main finding about the 25% of water restricted by membranes.

We are fully aware and agree with your concerns and wish to make the readers aware as well. We have removed the text “… which makes no assumptions about tissue microstructure.” In order to ensure that all readers understand the assumptions of the numerical inversion method and the interpretation of the distributions we have added to the main text:

“Data inversion methods used to obtain 1- and 2-D distributions of diffusion coefficients assume that the data attenuates exponentially with b. […] New development of nonparametric diffusion models may allow further interpretations.”

We utilize both raw signal and the distributions to estimate the restricted water fraction and have added the following text:

“At b x D_0_=6 (for which τ=0.63 ms and free water moves on average ld=D0τ=1.16 μm), freely diffusing water signal has attenuated to exp(-6)=0.0025 which is approximately the standard deviation of the noise. Signal at b x D0=6 (or the nearest data point) is used to define the restricted water fraction. Alternatively from diffusion coefficient distribution analysis, integrals of P(D/D0) on either side of D/D0=0.17 are heuristically used as measures of the free and restricted water fraction.”

“Both the signal attenuation and the distributions show that the mobility of a large portion of water is restricted to some degree during the diffusion encoding time. The restricted fraction quantified from distribution analysis is (mean ± SD) 0.23 ± 0.006. Alternatively from raw signal, the restricted fraction is 0.22 ± 0.002. Taking into account a few percent of the free water component being from aCSF bathing the sample, roughly 25% of the water in the tissue is restricted on the 1 ms timescale.”

10) Studying exchange using two diffusion encoding separated by a mixing time. This is the central part of the study, which – in a clever way – decouples from the problem associated with the previous part.

We are glad you appreciate this method towards quantifying exchange without the effects of restriction.

11) Studying the origin of restrictions using the substitution with deuterated water and the tissue delipidation – this part should have a high impact on the further development in the field.

We are glad you appreciate these experimental perturbations and our rigorous attempts to understand the origins of restriction.